# FluoEM, virtual labeling of axons in three-dimensional electron microscopy data for long-range connectomics

**Florian Drawitsch[1,2], Ali Karimi[1], Kevin M Boergens[1], Moritz Helmstaedter[1,2]***

[1]Department of Connectomics, Max Planck Institute for Brain Research, Frankfurt, Germany; [2]Donders Institute, Faculty of Science, Radboud University, Nijmegen, Netherlands

**Abstract** The labeling and identification of long-range axonal inputs from multiple sources within densely reconstructed electron microscopy (EM) datasets from mammalian brains has been notoriously difficult because of the limited color label space of EM. Here, we report FluoEM for the identification of multi-color fluorescently labeled axons in dense EM data without the need for artificial fiducial marks or chemical label conversion. The approach is based on correlated tissue imaging and computational matching of neurite reconstructions, amounting to a virtual color labeling of axons in dense EM circuit data. We show that the identification of fluorescent light-microscopically (LM) imaged axons in 3D EM data from mouse cortex is faithfully possible as soon as the EM dataset is about 40–50 µm in extent, relying on the unique trajectories of axons in dense mammalian neuropil. The method is exemplified for the identification of long-distance axonal input into layer 1 of the mouse cerebral cortex.

DOI: https://doi.org/10.7554/eLife.38976.001

## Introduction

The dense reconstruction of neuronal circuits has become an increasingly realistic goal in the neurosciences thanks to developments in large-scale 3D EM (*Denk and Horstmann, 2004*; *Bock et al., 2011*; *Hayworth et al., 2015*; *Kasthuri et al., 2015*; *Xu et al., 2017*) and circuit reconstruction techniques (*Saalfeld et al., 2009*; *Boergens et al., 2017*). First locally dense neuronal circuits have been mapped using these techniques (*Helmstaedter et al., 2013*; *Kasthuri et al., 2015*; *Berck et al., 2016*; *Wanner et al., 2016*). However, in mammalian nervous systems, any given local neuronal circuit receives numerous synaptic inputs from distant neuronal sources. In most layers of the mammalian cortex, for example, the fraction of distal input synapses in a dense local circuit is estimated to comprise 70–90% of all synapses (*Stepanyants et al., 2009*). Similarly, subcortical structures as the striatum or amygdala (*Figure 1a*) receive the vast majority of their inputs from distal projection sources. Uncovering the synaptic logic of such multi-source circuits together with the local dense neuronal connectivity requires the identification of multiple input sources in the same connectomic experiment. Since 3D EM is still limited to volumes that do not routinely encompass entire mammalian brains, the origin of a large fraction of input synapses remains thus unidentified in current high-resolution connectomic experiments in mammals.

At the same time, the labeling of multiple neuronal populations and their axonal projections by fluorescent dyes visible in LM is routine (*Livet et al., 2007*; *Mao et al., 2011*; *D'Souza et al., 2016*). Combining light-microscopic fluorescent labels in axons from multiple sources with large-scale 3D EM would be ideal. Such direct correlative light- and electron microscopy (CLEM) has been successful for high-resolution subcellular label localization (*Agronskaia et al., 2008*; *Murphy et al., 2011*; *Liv et al., 2013*) and synaptic identification (*Micheva and Smith, 2007*; *Rah et al., 2013*), but

***For correspondence:**
mh@brain.mpg.de

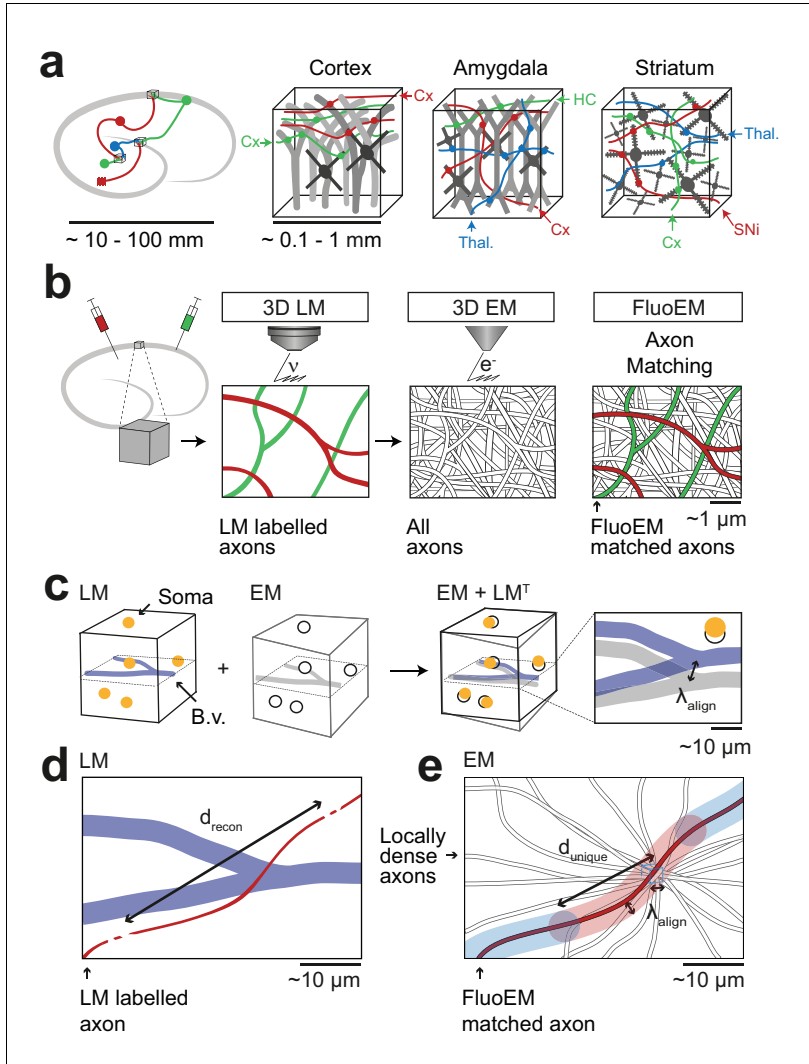

**Figure 1.** FluoEM applications and concept. (**a**) Most local circuits in mammalian brains can be mapped densely by modern 3D EM (gray neurons), but the source of a majority of the relevant input axons remain unidentified (colored), because the projections arise from distal brain regions (see sketch on the left). Examples show upper layers of cortex, Amygdala and Striatum with dominant input from other cortices (Cx), Thalamus (Thal.), Hippocampus (HC), Substantia Nigra (SNi). Deciphering the connectomic logic of these inputs within a densely imaged neuronal circuit requires parallel axonal labels in a single EM experiment. Mapping color-encoded information on multiple axonal origins from LM data into 3D EM data useable for connectomic reconstruction in mammalian nervous tissue is the goal of the presented method. (**b**) A brain tissue sample containing fluorescently labeled axons is volume imaged first using confocal laser scanning microscopy (LM) and then using 3D electron microscopy (EM). The fluorescently labeled axons (red and green) contained in the LM dataset are then matched to the corresponding axons in the EM dataset (black and white) solely based on structural constraints (see b-d), without chemical label conversion or artificial fiducial marks. (**c**) Coarse LM-EM volume registration based on intrinsic fiducials such as blood vessels (blue) and somata (yellow) can be routinely achieved at a registration precision $\lambda_{align}$ of about 5 –10 µm (***Bock et al., 2011***; ***Briggman et al., 2011***). (**d**) The path length $d_{recon}$ over which fluorescently labeled axons (green and red) in a volume LM dataset are clearly reconstructable depends mostly on the labeling density in the respective fluorescence channel. Blue, blood vessels (B.v.). (**e**) Axons (grey) traversing a common bounding box (blue) of extent $\lambda_{align}$ become increasingly distinguishable with increasing distance from the common origin. After a distance $d_{unique}$, a given axonal trajectory is unique, that is no other axon from the center bounding box is still closer than $\lambda_{align}$. Since 3D EM allows the reconstruction of all axons traversing a given bounding box, $d_{unique}$ can be determined (see ***Figure 2***). If axons can be reconstructed at the LM level for a path length $d_{recon}$ (c) that is similar to or larger than the typical axonal uniqueness length $d_{unique}$ (d), the FluoEM approach can in principle be successful.

DOI: https://doi.org/10.7554/eLife.38976.002

can so far not be combined with 3D EM at both resolution and scale allowing for dense large-scale circuit mapping in mammalian neuropil.

Similarly, en-bloc immunolabeling in conjunction with EM is limited to thin tissue slices (*Faulk and Taylor, 1971*; *Pallotto et al., 2015*). Approaches to interlace EM and LM imaging on alternating tissue slices, though successfully applied in invertebrates (*Shahidi et al., 2015*), impede dense EM-based circuit reconstruction in mammalian nervous tissue. Rather, methods to obtain electron-dense labels in a subset of neurons have been developed over the decades based on induced axonal degeneration (*Gray and Hamlyn, 1962*; *Colonnier, 1964*; *White, 1978*), introduction of HRP (*Hersch and White, 1981*; *Hamos et al., 1985*; *Horikawa and Armstrong, 1988*; *Anderson et al., 1994a*; *Anderson et al., 1994b*; *Markram et al., 1997*; *da Costa and Martin, 2011*) or fluorophores (*Maranto, 1982*; *Grabenbauer et al., 2005*; *Knott et al., 2009*) into a subset of neurons which could be utilized to create electron contrast via enzymatic or photo-oxidative DAB polymerization. More recent improvements enabled the genetically targeted expression of electron-dense labels in diverse cellular compartments, potentially allowing to encode multiple label classes in a single EM experiment (*Shu et al., 2011*; *Martell et al., 2012*; *Atasoy et al., 2014*; *Lam et al., 2015*; *Joesch et al., 2016*). However, generating multiple electron-dense label classes in the very same experiment while ensuring ultrastructural preservation is challenging and to our knowledge has been accomplished in *Drosophila* (*Lin et al., 2016*) but so far not in mammals. Furthermore, the number of cellular compartments usable for such axonal identification may be rather limited.

Methods to directly co-register light microscopic labels in axons and dendrites with 3D EM data (*Bishop et al., 2011*; *Maco et al., 2013*; *Gala et al., 2017*) have been restricted to local volumes of up to 10 μm on a side, using LM-burnt fiducial marks visible in EM; these do not scale to the large volumes required for neuronal circuit reconstruction (sized hundreds of micrometers per dimension, that is at least 1000-fold larger). Direct co-alignment between fluorescently labeled cell bodies and EM data has been routinely applied (*Bock et al., 2011*; *Briggman et al., 2011*; *Lee et al., 2016*; *Tsang et al., 2018*), but provides LM-to-EM alignment only at a coarse scale that does not allow for the identification of single axons.

Apart from the experimental limitations currently hampering the use of multi-color labels in large 3D EM data, current computational approaches are challenged when attempting submicron registration precision of neurites in volumes required for connectomics ($\sim10^6$ μm$^3$ or larger). Previously proposed multi-modal graph-based neurite registration (*Serradell et al., 2012*; *Fua and Knott, 2015*; *Serradell et al., 2015*) can in principle be used for LM-to-EM morphological matching, but did not yet scale to graphs of thousands or tens of thousands of nodes as required for connectomics.

Thus, a method that could directly employ the full multi-color space of LM in large 3D tissue samples while enabling dense EM-based circuit reconstruction would be ideal. Here, we report FluoEM, a set of experimental and computational tools to achieve this goal. Instead of directly attempting alignment of LM-labeled axons to the EM dataset at the nanometer scale, we exploit the availability of locally dense neurite reconstructions in current volume EM. Here, the alignment problem can be re-formulated as identifying the most likely axon (out of all axons, reconstructed in EM) that best explains an LM-imaged axonal fluorescence signal. We show that this is a well-constrained problem based on the actual geometry of axonal fibers in nerve tissue from mouse cerebral cortex. We report the experimental methods allowing for subsequent correlative imaging of samples in LM and EM and the computational tools to register the labeled LM axons to their EM correspondences. We exemplify our method for long-range inputs to layer 1 of mouse cortex.

## Results

The FluoEM workflow (*Figure 1b*) comprised the following steps: acquisition of a 3D fluorescence-LM dataset from a given piece of tissue; 3D imaging of that very same piece of tissue in the electron microscope; reconstruction of the fluorescently labeled axons in the LM data; reconstruction of locally all axons in small subvolumes of the EM data; computational determination of the most likely EM axons that explain the LM signal.

In a first step, we used blood vessels as intrinsic fiducials in the neuropil to obtain a coarse co-registration between LM and EM (*Figure 1b*). Coarse LM-EM registration based on cell bodies and blood vessels has been successfully performed before (*Bock et al., 2011*; *Briggman et al., 2011*) and is expected to yield an alignment precision $\lambda_{align}$ of about 5–10 μm (*Figure 1c*). We next had to

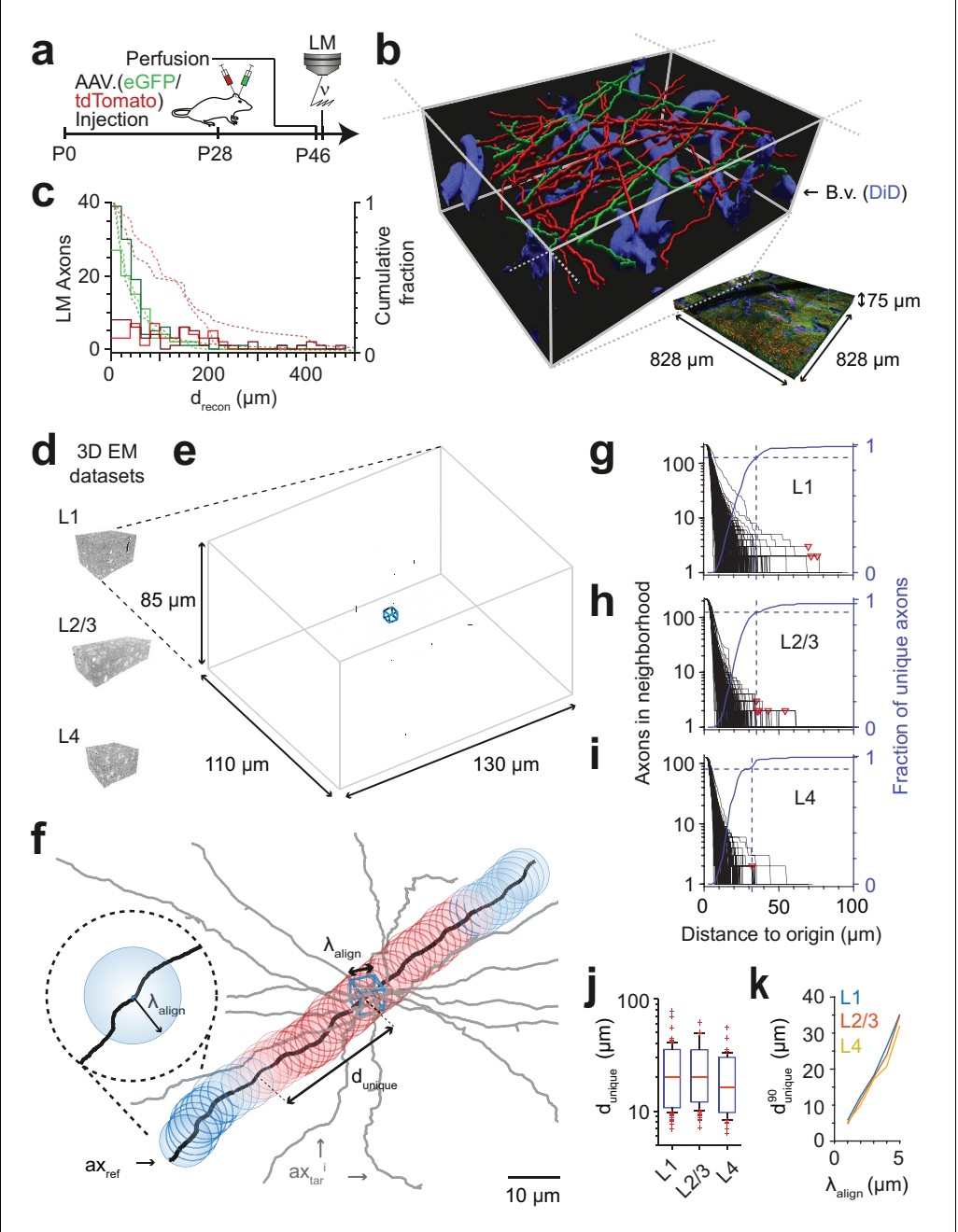

**Figure 2.** FluoEM proof-of-principle measurements. (**a–c**) Measurement of the length $d_{recon}$ over which axons in LM fluorescence data are faithfully reconstructable (see Fig. 1d). (**a**) Experimental timeline: Injection of a fluorescent-protein (FP) expressing adeno-associated virus at postnatal day 28 (AAV.eGFP into M1 cortex, AAV.tdTomato into S2 cortex of a wild-type C57BL/6j mouse); transcardial perfusion, fixation, sample extraction from cortical L1 at postnatal day 46 followed by confocal volume imaging. (**b**) Rendering of a subvolume of the three-channel confocal image stack showing axonal long-range projections from M1 cortex (green), S2 cortex (red) as well as stained blood vessels (blue). (**c**) Histogram of the reconstructable pathlength ($d_{recon}$) for the fluorescently stained long-range axons (red, green) in the example LM dataset (b) based on two independent expert annotations (light and dark colors; median $d_{recon}$ of 28 µm and 33 µm (green channel, n = 115 and 88) and 88 µm and 107 µm (red channel, n = 56 and 47) for the two annotators, respectively). Dashed lines: cumulative axon fraction per annotator and channel. Note that annotators were explicitly asked to stop reconstructions whenever any ambiguity or uncertainty occurred, thus biasing these reconstructions to shorter length. (**d–k**), EM-based measurements of the length $d_{unique}$ over which the trajectory of axons becomes unique in several cortical layers. (**d**) Renderings of three

*Figure 2 continued*

3D EM datasets in layers 1, 2/3 and 4 of mouse cortex. (**e**) Skeleton reconstructions in the L1 3D EM dataset of all 220 axons (black) that traversed a center bounding box (blue) of edge length $\lambda_{align}$ = 5 µm. (**f**) Determination of the uniqueness length $d_{unique}$ for a given axon (black) by counting the number of other axons (gray) that were within the same seeding volume (blue box) and remain within a distance of no more than $\lambda_{align}$ from the reference axon for increasing distances $d$ from the center box. $d_{unique}$ was defined as the Euclidean distance from the center box at which no other axon persistently was at less than $\lambda_{align}$ distance from the reference axon (red: at least one neighboring axon, blue: no more neighboring axon). (**g**) Axonal uniqueness length $d_{unique}$ in cortical L1 for all n = 220 axons from the center bounding box (see e). Number of neighboring axons for each axon (black) and the combined fraction of unique axons (blue) over Euclidean distance from the bounding box center. Note that at $d_{unique}^{90}$= 35 µm the trajectory of 90% of all axons has become unique (dashed blue line). For three of 220 axons, one or two other axons remained in $\lambda_{align}$ proximity within the dataset (red triangles). (**h,i**) Axonal uniqueness length in two additional 3D EM datasets from L2/3 (h, n = 207 axons) and L4 (i, n = 128 axons), labels as as in g. (**j**) Box plot of uniqueness length per axon for cortical layers 1-4 (from g-i). Boxes indicate (10th, 90th)-percentiles, whiskers indicate (5th, 95th)-percentiles, red crosses indicate outliers. (**k**) Effect of the coarse alignment precision $\lambda_{align}$ on axonal uniqueness length $d_{unique}^{90}$ (resampled for smaller $\lambda_{align}$ = 1, 2, 3, 4 µm, see Materials and methods).

DOI: https://doi.org/10.7554/eLife.38976.003

The following source data and figure supplement are available for figure 2:

**Source data 1.** Axonal uniqueness analysis for cortical layer 1 (*Figure 2g*).
DOI: https://doi.org/10.7554/eLife.38976.005

**Source data 2.** Axonal uniqueness analysis for cortical layer 2/3 (*Figure 2h*).
DOI: https://doi.org/10.7554/eLife.38976.006

**Source data 3.** Axonal uniqueness analysis for cortical layer 4 (*Figure 2i*).
DOI: https://doi.org/10.7554/eLife.38976.007

**Source data 4.** Axonal uniqueness summary boxplot (*Figure 2j*).
DOI: https://doi.org/10.7554/eLife.38976.008

**Source data 5.** Axonal uniqueness dependence on alignment error (*Figure 2k*).
DOI: https://doi.org/10.7554/eLife.38976.009

**Figure supplement 1.** Axonal uniqueness length $d_{unique}$ for different cortical layers corrected for numbers of contained axons.
DOI: https://doi.org/10.7554/eLife.38976.004

determine the typical length of faithful 3D axon reconstruction $d_{recon}$ in fluorescence data (*Figure 1d*). Then, the problem of matching EM axons to LM data can be phrased as follows (*Figure 1e*): Given all axons that traverse a cube of edge length $\lambda_{align}$ in the EM data, will the trajectory of these axons become unique on a length scale $d_{unique}$ that is smaller than or comparable to the LM-reconstruction length $d_{recon}$: $d_{unique}$ $(\lambda_{align}) \leq d_{recon}$? In other words: What is the travel length $d_{unique}$ after which all axons from a center cube of size $\lambda_{align}$ will have no other of these axons closer than $\lambda_{align}$?

In order to understand whether this approach could work, we had to measure $d_{recon}$ in LM data with realistic labeling density and determine $d_{unique}$ for a realistic $\lambda_{align}$ in EM data. These measurements are reported next, followed by a proof-of-principle co-registration experiment and results from the virtual labeling of axons in EM data.

## Reconstruction length and axon uniqueness

We first performed a fluorescence imaging experiment (*Figure 2a–c*) in which we injected adeno-associated viruses expressing fluorescent markers into the primary motor cortex (M1; virus expressing the green marker eGFP) and the secondary somatosensory cortex (S2; virus expressing the red marker tdTomato) of a 28-day-old mouse, followed by transcardial perfusion and tissue extraction at 46 days of age. Using a confocal laser scanning microscope, we then acquired a dataset sized (828 × 828 × 75) µm³ at a voxel size of (104 × 104 × 444) nm³ from layer 1 of a part of cortex in which both of these projections converged (located to lateral parietal association cortex (LPtA), *Figure 2b*). We then 3D-reconstructed all axons traversing a (40 × 40 × 48) µm³ bounding box in this LM image dataset using our annotation tool webKnossos (*Boergens et al., 2017*) and measured the length over which the axons were faithfully reconstructable as judged by expert annotators (for

this we asked the annotators to stop reconstructing whenever they encountered a location of any ambiguity, the loss of axon continuity or an unclear branching). The reconstruction length $d_{recon}$ per axon was $70 \pm 44$ µm (mean ± s.d., n = 46 axons) for the green and $134 \pm 62$ µm (n = 42 axons) for the red fluorescence channel. Ninety-five percent of axons were reconstructable above 20 µm length, some for up to 200 µm (*Figure 2c*, independent reconstruction by two experts).

We then measured the length $d_{unique}$ over which the trajectory of axons in cortex becomes unique, assuming a macroscopic alignment precision of $\lambda_{align}$ = 5 µm (*Figure 2d–k*). For this, we acquired a 3D EM dataset sized (130 x 110 x 85) µm$^3$ at a voxel size of (11.24 x 11.24 x 30) nm$^3$ from layer 1 of LPtA cortex (*Figure 2d,e*) using SBEM (*Denk and Horstmann, 2004*) and reconstructed all n = 220 axons that traversed a cubic bounding box of size (5 x 5 x 5) µm$^3$ in the center of the dataset (*Figure 2e*, blue box). We then analyzed for each axon how many of the other 219 axons were still within $\lambda_{align}$ proximity at a Euclidean distance $d$ from the center cube (*Figure 2f*). For a given axon, the uniqueness length was then computed as the distance after which no other axon was in the $\lambda_{align}$-surround. *Figure 2g* shows the number of remaining axons after distance $d$ for a dense reconstruction of axons in L1 of mouse cortex. The uniqueness length per axon was $22 \pm 10$ µm (mean ± s.d., range 7 – 78 µm, n = 220 axons); after 31, 35, 41 and 69 µm, 85%, 90%, 95% and 98% of axons had no other axon in their proximity, respectively. We repeated this analysis in two additional 3D EM datasets obtained from layers 2/3 and 4 of mouse somatosensory cortex (datasets 2012-09-28_ex145_07x2 and 2012-11-23_ex144_st08x2, KM Boergens & MH, unpublished, Figures 2h,i). Here, uniqueness lengths per axon were $21 \pm 9$ µm (mean ± s.d., range 7 – 62 µm, n = 214, L2/3) and $17 \pm 8$ µm (mean ± s.d., range 6 – 55 µm, n = 128, L4), with 90% of axons unique after $d_{unique}^{90}$= 35 µmin L2/3 and $d_{unique}^{90}$= 32 µm in L4 (Fig. 2j, $d_{unique}^{90}$was indistinguishable between layers 1, 2/3 and 4, *Figure 2—figure supplement 1*).

Importantly, when comparing $d_{unique}^{90}$ in layer 1 as determined in 3D EM data to the reconstruction lengths $d_{recon}$ obtained in the fluorescence datasets (*Figure 2c*), we find that about 45% (green channel) and about 80% (red channel) of the faithfully LM-reconstructed axonal stretches were at least of length $d_{unique}^{90}$. Since for a first LM-EM co-registration a set of the longest LM-reconstructable axons can be used, these data strongly suggested that the FluoEM approach for matching-based co-alignment between LM and EM reconstructed axons could in principle be successful. Furthermore, we expected the uniqueness length $d_{unique}^{90}$ to depend on the macroscopic alignment precision $\lambda_{align}$ achieved by the blood-vessel-based coarse pre-registration of the LM and EM datasets (*Figure 2k*). When re-analyzing our dense axonal reconstructions in the 3D EM dataset from L1 for smaller $\lambda_{align}$, we find that the uniqueness length strongly decreased for smaller $\lambda_{align}$ (to $d_{unique}^{90} = 5$ µm at $\lambda_{align} = 1$ µm; $d_{unique}^{90} = d_{recon}^{90}$(eGFP) = 11 µm at $\lambda_{align} = 2$ µm; $d_{unique}^{90} = d_{recon}^{90}$(tdTomato) = 20 µm at $\lambda_{align} = 3$ µm).

## Correlated LM and 3D EM dataset acquisition

We then set out to perform correlated 3D LM-EM imaging for the application of FluoEM (*Figure 3*). For this, we continued the experiment described above (*Figure 2a*) and acquired an additional fluorescence channel to image blood vessels (during perfusion, the rinsing solution was supplemented with DiD, a lipophilic dye for blood vessel staining with fluorescence in the far red spectrum, around 650 nm). Then we applied a slightly modified version of our standard volume EM staining protocol (*Hua et al., 2015*) to the tissue block (see Materials and methods), embedded and mounted the sample and imaged it using SBEM (*Denk and Horstmann, 2004*) (*Figure 3a*). We first acquired EM images with a large field of view (dataset dimensions (1500 × 1000 × 10) µm$^3$) at lower resolution of (270 × 270 × 200) nm$^3$ to identify the location of the LM-imaged dataset relative to the EM imaged sample (*Figure 3b,c*). For this, we used the pattern of superficial blood vessels as coarse alignment fiducials (*Figure 3d,e*). We then acquired a high-resolution 3D EM dataset sized (130 × 110 × 85) µm$^3$ at a voxel size of (11.24 × 11.24 × 30) nm$^3$ in a region previously imaged using LM with suitable fluorescent labeling density (*Figure 3c,g*). In both the high-resolution EM dataset and the corresponding LM volume we then reconstructed the blood vessels (*Figure 3h,i*) and annotated their branch points as control points for fitting a coarse affine transformation $AT_{BV}$ from the LM to the high-resolution EM dataset (*Figure 3h–n*, n = 7 control points). To estimate the precision of this coarse alignment, we measured its residuals (*Figure 3o*; $2.7 \pm 1.1$ µm, mean ± s.d.,

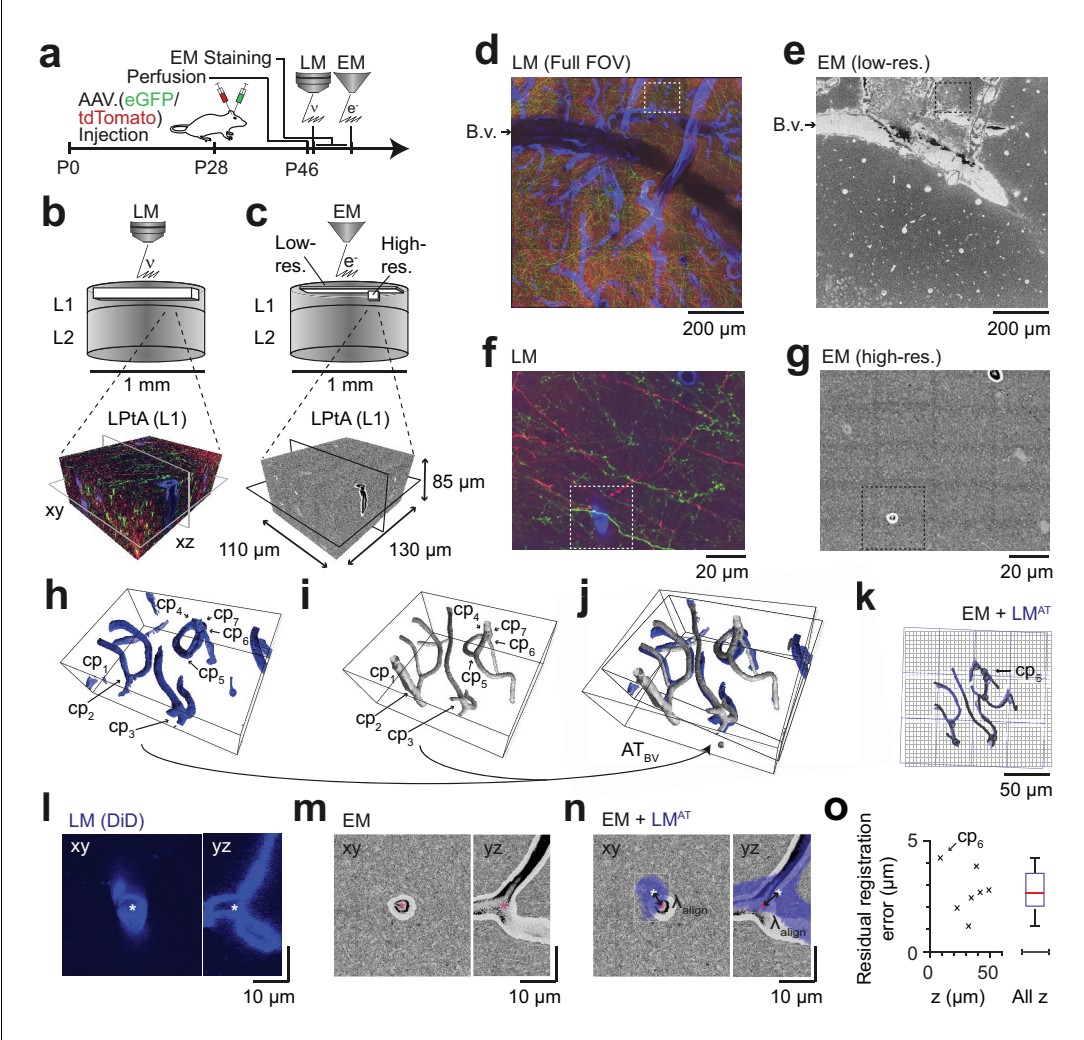

**Figure 3.** Coarse LM-EM registration. (**a**) Experimental timeline for FluoEM experiment (continued from Fig. 2a). After LM imaging was concluded the sample was immediately stained for EM and embedded, followed by 3D EM imaging. (**b,c**) Sketch of sequentially acquired 3D LM and EM datasets from cortical layer 1 (L1). Fluorescence channels: green (axonal projections from M1 cortex, eGFP); red (axonal projections from S2 cortex, tdTomato); blue (blood vessels, DiD). Volume and voxel size of EM datasets: (1500 x 1000 x 10) $\mu m^3$ and (700 x 700 x 200) $nm^3$ (low-resolution EM dataset), (130 x 110 x 85) $\mu m^3$ and (11.24 x 11.24 x 30) $nm^3$ (high-resolution EM dataset). (**d,e**) Overview images of the LM (d, https://wklink.org/5974) and low-resolution EM (e, https://wklink.org/4610) datasets. Dashed rectangle indicates the position of the high-resolution EM dataset (g, see c for relative positions of datasets). B.v., blood vessel. Note the corresponding pattern of surface vessels in d,e. (**f,g**) High-resolution images at corresponding locations in LM (f, https://wklink.org/1204) and EM (g, https://wklink.org/5386). Dashed rectangles indicate regions shown in l-n. (**h,i**), Rendering of blood vessel segmentations in LM (h) and EM (i) and their characteristic bifurcations used as control point pairs ($cp_{1-7}$) to constrain a coarse affine transformation ($AT_{BV}$). (**j**) Overlay of the EM and $LM^{AT}$ blood vessel segmentations registered using $AT_{BV}$. (**k**) Registered blood vessels (lines) and control points (circles) overlaid with EM (black) and transformed LM (blue) coordinate grids. (**l–n**) LM (l) and EM (m) image planes (xy) and reslice (yz) showing an exemplary blood vessel bifurcation (asterisk indicates $cp_6$, see h-k) and the overlay (n) of the affine transformed LM blood vessel segmentation with EM. Arrows indicate registration residual, a measure of the coarse alignment precision $\lambda_{align}$. (**o**) Alignment precision $\lambda_{align}$ reported as residuals of control point alignment along dataset depth (denoted with z) (mean: 2.7 ± 1.1 $\mu m$, median: 2.7 $\mu m$) after affine transformation $AT_{BV}$.

DOI: https://doi.org/10.7554/eLife.38976.010

The following source data is available for figure 3:

**Source data 1.** Blood vessel based affine registration error (**Figure 3o**).

DOI: https://doi.org/10.7554/eLife.38976.011

n = 7, median = 2.7 $\mu m$). Since this coarse alignment precision was well in the range previously assumed for the analysis of axonal uniqueness ($\lambda_{align} \approx$ 5 $\mu m$, **Figure 3g–k**), we then attempted a first LM-to-EM matching.

## LM-to-EM axon matching

We chose an axon next to a blood vessel in the LM data (*Figure 4a*). The axon was skeleton-reconstructed at the LM level (*Figure 4b*) using webKnossos (*Boergens et al., 2017*). We then transformed this LM reconstruction (path length of 105 μm) to the EM dataset using the coarse transformation $AT_{BV}$ based on blood vessel correspondence as determined before (*Figure 4c*). Evidently, based solely on this coarse transformation, an identification of the corresponding axon at the EM level was impossible (*Figure 4d*). Rather, we then defined a cubic region sized $(5 \ \mu m)^3$ in the EM data next to the blood vessel at which the LM reconstruction had been seeded (*Figure 4e,f*) and reconstructed all axons that traversed that region in the EM dataset (*Figure 4g*, n = 193 axons). We then used a spherical search kernel with 5 μm radius (*Figure 2f*) to determine which of the 193 EM-reconstructed axons were within that surround of the transformed LM-reconstructed axon along all of its trajectory (*Figure 4h,i*). Beyond a distance of 42 μm along the transformed LM axon, only one candidate EM-reconstructed axon remained persistently within the search kernel (*Figure 4i*). So, in fact, for this example fluorescent axon, we had found a corresponding axon at the EM level that matched its trajectory.

But how could we be sure that this EM axon was the one that had given rise to the fluorescence signal at the LM level? We first investigated the detailed morphology of this axon and identified axonal varicosities in the EM data. We then reconstructed the varicosities visible in the LM data and overlaid both (*Figure 4j*). The similarity was visually striking (distance between LM and EM varicosities, 0.7 μm ± 0.6 (mean ± s.d., n = 11); distance between LM and random EM varicosity distributions: 4.9 μm ± 2.1, (mean ± s.d., n = 11 varicosities, n = $10^5$ draws, p < $10^{-5}$, Randomization test, *Figure 4—figure supplement 1*)). With this, we had found the EM axon that most likely explained the fluorescence signal reconstructed in the LM dataset (*Figure 4k*).

We then continued to match other fluorescent axons given this first matched axon (*Figure 5*). For this, we chose an axon in the LM data in proximity to the previously matched axon and selected an LM varicosity close to the axonal apposition as a seed point for the correspondence search in the EM data (*Figure 5a*). Since each matched axon provided additional registration constraints beyond the initial blood-vessel-based constraints, we could now use each matched axon to further refine an elastic-free-form LM-to-EM transformation using the matched varicosities, branchpoints and other prominent structural features of the axons seen in both LM and EM (*Figure 4b,k*). The iteratively refined registration together with the usage of varicosities as search criteria allowed us to substantially reduce the work load for matching subsequent fluorescent axons to their EM counterparts (*Figure 5b–e*; in this example 34 axons were within a $(2 \times 2 \times 3) \ \mu m^3$ search volume around the LM-varicosity, but only six of these had a varicosity). While the first match involved the reconstruction of 193 axons, the subsequent matches were successful after only several or even one reconstruction attempt and a few minutes of reconstruction time (*Figure 5e*). The fact that all subsequent matches were successful (see varicosity correspondence as shown in *Figure 5f*) implies that the previous matches were correct (since the probability of finding a correctly matched axon by chance after several chance matches is virtually zero). Using this strategy, we matched a total of 38 axons (27 in the red fluorescence channel, 11 in the green channel), using a total of 284 control points (7.5 ± 3.8 cps per axon (mean ± s.d), *Figure 5g*).

To quantify the improved registration precision obtained from the iteratively constrained LM-to-EM axon matches, we measured the residual registration error of both affine and free-form registration (constrained by a random subset of 250 control points) on 30 other control points. Compared to the affine registration, the residual registration error was reduced by one third from 725 ± 408 nm (mean ± s.d) to 487 ± 290 nm (mean ± s.d, *Figure 5h*; *Figure 5—figure supplement 1*), thus about six-fold smaller than in the blood vessel based coarse pre-alignment obtained before (cf. *Figure 3o*; see *Figure 5i* for dependence on control point number).

To test whether we could perform a close-to-complete matching of LM-to-EM reconstructions for a certain depth range of the fluorescence dataset we matched all clearly visible fluorescently labelled axons at a depth of 17–20 μm in the red (tdTomato) channel (n = 27 axons, total path length of 3 mm). We then computed a segmentation of the LM data and measured the fraction of the segmentation voxels overlapping with the EM-matched reconstructions (*Figure 5j–n*). We found that in fact about 90% (86 ± 3%, mean ± s.d, n = 7 LM image planes) of the fluorescent voxels were explained by the matched EM axons in this depth range (*Figure 5o*).

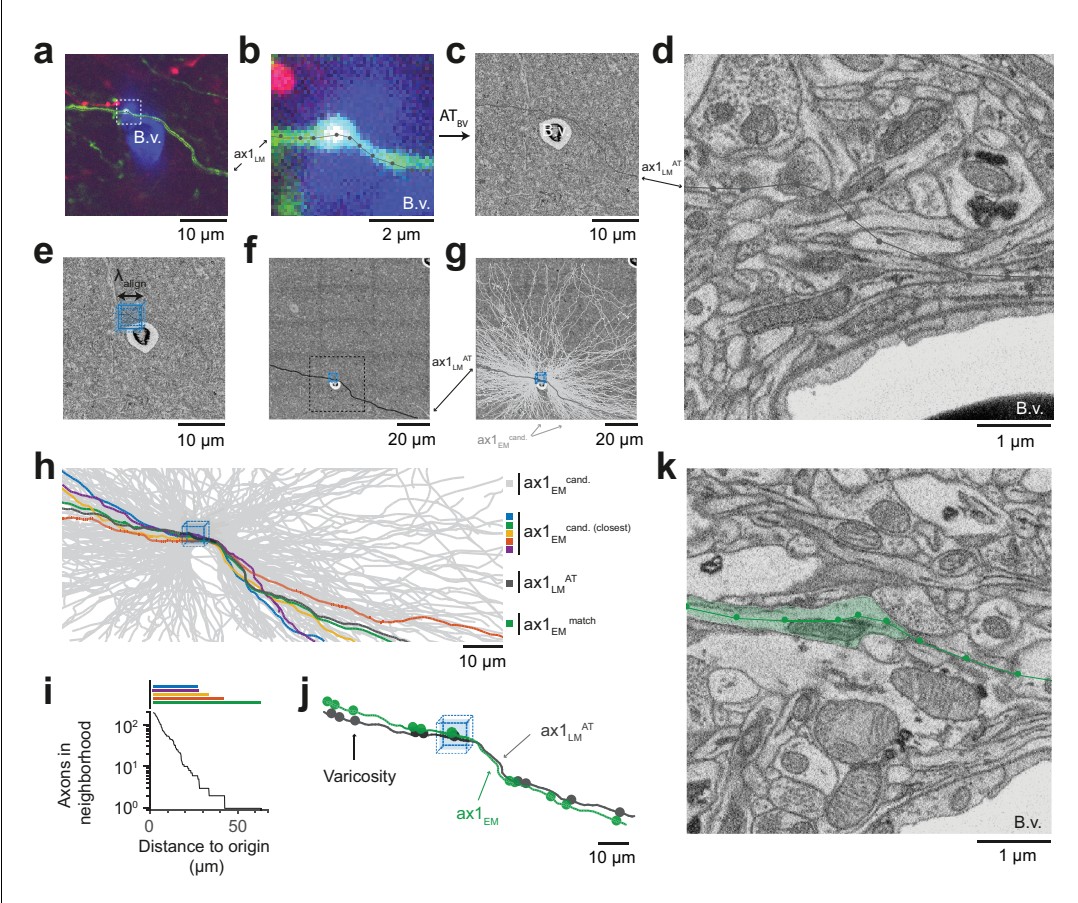

**Figure 4.** Initial LM-to-EM axon matching. (**a,b**) LM image plane showing an axon (green) overlaid with its LM-based skeleton reconstruction (ax1$_{LM}$, black) and a nearby blood vessel (B.v., blue) bifurcation (a, https://wklink.org/1399). Dashed rectangle, region magnified in (b, https://wklink.org/5406). (**c,d**) EM image plane showing the blood vessel (B.v.) corresponding to the vessel shown in (a,b) overlaid with the coarse affine transformation (see *Figure 3*) of the LM-based axon reconstruction ax1$_{LM}^{AT}$; magnified in d. Note that ax1$_{LM}^{AT}$ does not match the ultrastructural features at high-resolution EM (**d**) due to the limited precision of AT$_{BV}$. (**e,f**), EM plane overlaid with ax1$_{LM}^{AT}$ (as shown in c) with bounding box for dense axon reconstruction (light blue) of edge length $\lambda_{align}$ = 5 μm; demagnified in f, dashed rectangle denotes region shown in e. (**g**) Skeleton reconstructions of all n = 193 ax1$_{EM}$ candidates (light grey) traversing the search bounding box (as in e-f), https://wklink.org/5897 (**h**) Reconstructed ax1$_{EM}$ candidates (light grey) and the transformed fluorescent axon skeleton ax1$_{LM}^{AT}$ (black). Five candidate EM reconstructed axons with the most similar trajectories as ax1$_{LM}^{AT}$ highlighted in different colors. (**i**) Number of EM-reconstructed axons persistently within the $\lambda_{align}$ neighborhood of the LM-based template axon ax1$_{LM}^{AT}$. Note that at a distance of 43 μm from the seed center (e-h), only one possible ax1$_{EM}$ candidate (green) remains. (**j**) Overlay of ax1$_{LM}^{AT}$ and the most likely ax1$_{EM}$ candidate reconstruction. Axonal varicosities (filled circles) were independently annotated in LM and EM. Note the strong resemblance of the varicosity arrangements indicating that ax1$_{EM}$ is the correctly matched EM equivalent of the LM signal (see *Figure 4—figure supplement 1*) for statistical analysis). (**k**) EM plane with overlaid ax1$_{EM}$ skeleton (dark green) and volume annotation of the corresponding EM axon ultrastructure (transparent green), https://wklink.org/3268.

DOI: https://doi.org/10.7554/eLife.38976.012

The following source data and figure supplement are available for figure 4:

**Source data 1.** Axonal uniqueness analysis for initial axon match (*Figure 4i*).
DOI: https://doi.org/10.7554/eLife.38976.014
**Figure supplement 1.** Similarity of matched varicosity patterns between LM and EM.
DOI: https://doi.org/10.7554/eLife.38976.013

In summary, investing a total of 195 work hours (178.8 hr for the first axon; 16.6 hr for the remaining axons) we obtained the EM equivalents of 38 fluorescently labeled axons in two fluorescence channels (total axonal path length of 4.6 mm). The total reported time included LM-to-EM matching (181.1 hr), full axon reconstruction in EM (4.8 hr) and control point placement in LM and EM (9.5 hr). Since the matching of the first axon consumed most invested time, we checked whether by using

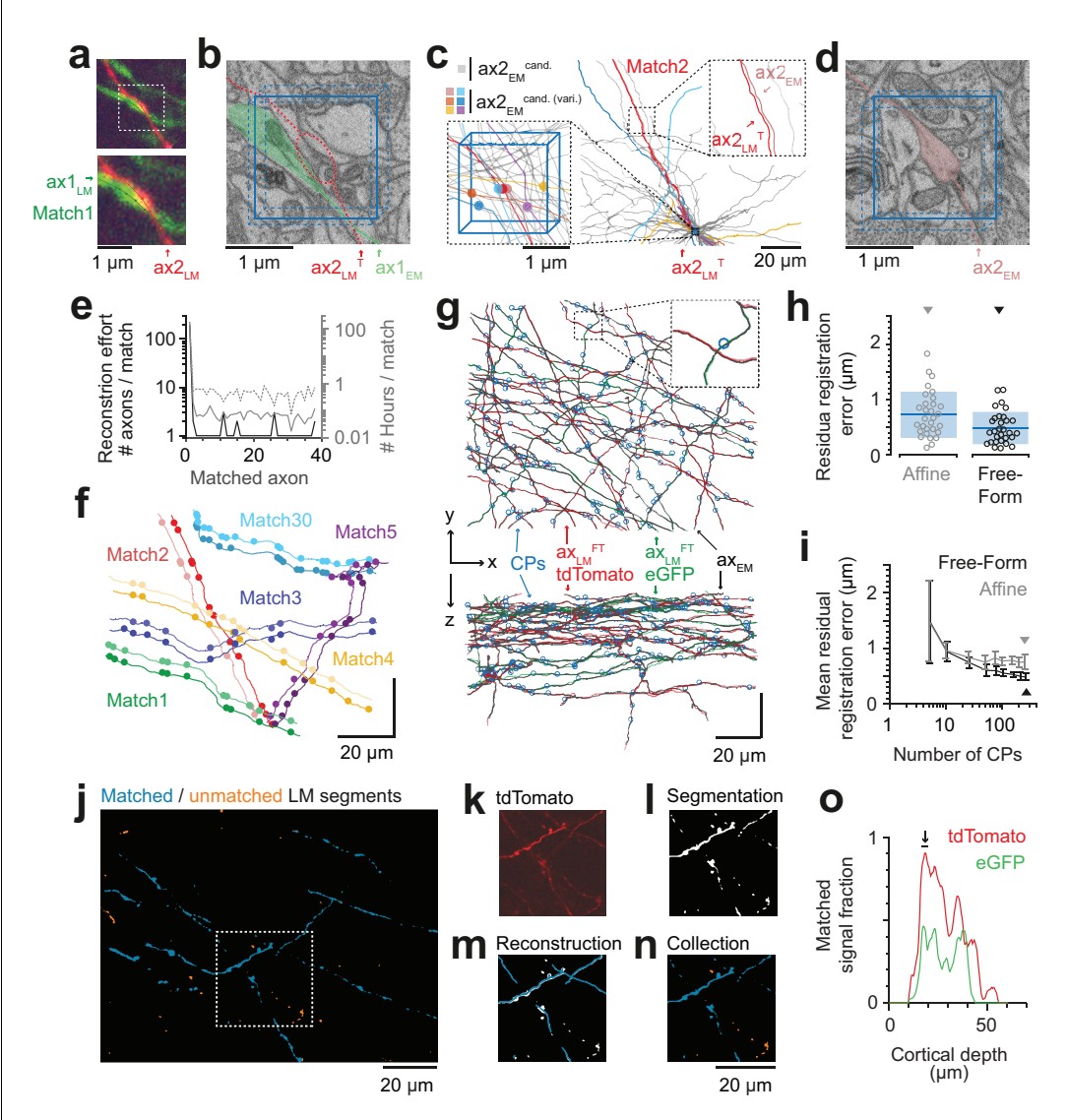

**Figure 5.** Iterative LM-to-EM axon matching. (a) Initial axon match $ax1_{LM}$ and nearby axon $ax2_{LM}$ with a varicosity in the proximity, both reconstructed in LM, https://wklink.org/4334. (b) Matched axon $ax1_{EM}$ (light green) and trajectory of axon 2 ($ax2_{LM}^T$) reconstructed at LM and transformed to EM (dashed red line) using a transformation T that was constrained using structural features of the initially matched ax1 (eight control points), https://wklink.org/3305. (c) Search volume sized ($2 \times 2 \times 3$) $\mu m^3$ (blue box) around the varicosity of $ax2_{LM}^T$ (see a-b) containing 34 candidate axons (gray) but only six with candidate varicosities (colored circles). One candidate axon had a similar trajectory as $ax2_{LM}^T$ (right inset), identifying it as the correct match (see also varicosity pattern in f). (d) Varicosity in EM of axon $ax2_{EM}$ (red shading, see search template in b), https://wklink.org/3820. (e) Reconstruction effort in terms of number of reconstructed axons (black) and time (grey) required to perform 38 iterative LM-to-EM axon matches. Total effort including full EM-based axon reconstruction and control point placement in LM and EM (dashed grey). (f) Overlay of EM (light colors) and affine transformed LM (dark colors) axon reconstructions of six exemplary matched axon pairs with locations of axonal varicosities (independently reconstructed at LM and EM level, respectively). Note the similarity of axon trajectory and varicosity positions, also for the 30th matched axon. EM axons were offset by 6 µm for visibility, see g for actual overlap quality. (g) Overlay of EM-reconstructed axon skeletons (black) with the matched LM-reconstructed skeletons transformed using a free-form transformation iteratively constrained by control points (CPs, blue circles) obtained from each consecutive axon match (shown transformation used a random subset of 250 (of 284) CPs). (h) Residual registration error of the match shown in g computed as the Euclidean distances || $cp_{EM} - cp_{LM}^{FT}$ || between n = 30 randomly picked CP pairs that had not been previously used to constrain the registration. Sample mean (blue line) and standard deviation (blue shading). (i) Average residual registration error (mean ± s.d., computed as in h) in dependence of the number of randomly chosen control points (CPs) used to constrain the transformation (n = 10 bootstrapped CP sets, each). (j–n) Locally complete LM-to-EM matching of fluorescently labeled axons. One example image plane of LM dataset (at 18 µm depth) shown with EM-matched fluorescence signal fraction (blue) and yet unmatched segments (orange). To compute the matched fluorescence signal fraction, the raw LM data (k) was binarized and segmented (l, see Materials and methods), overlaid with the EM-matched LM axon skeleton reconstructions (m) and only those segments overlapping with skeletons were

*Figure 5 continued on next page*

*Figure 5 continued*

counted as matched (n). (o) Matched signal fraction (in fraction of matched voxels) over dataset depth. For a range of 17–20 µm depth (black line, arrow), about 90% of the fluorescent voxels were explained by matched EM axons.

DOI: https://doi.org/10.7554/eLife.38976.015

The following source data and figure supplement are available for figure 5:

**Source data 1.** Reconstruction effort (*Figure 5e*).
DOI: https://doi.org/10.7554/eLife.38976.017
**Source data 2** Axon based affine and free-form registration error (*Figure 5h*).
DOI: https://doi.org/10.7554/eLife.38976.018
**Source data 3.** Averaged axon based affine and free-form registration error depending on control point numbers (*Figure 5i*).
DOI: https://doi.org/10.7554/eLife.38976.019
**Source data 4.** Matched LM signal fraction (*Figure 5o*).
DOI: https://doi.org/10.7554/eLife.38976.020
**Figure supplement 1.** Deformations of axons in LM vs EM data.
DOI: https://doi.org/10.7554/eLife.38976.016

the distribution of varicosities along EM and LM axons also for the initial match the time investment could be further reduced. In fact, when asking a second independent expert to perform the first LM-to-EM axonal match by also using varicosities as a search criterion, the invested total time was only 30.3 min for the initial match (7.5 min for EM-to-LM matching, 6.8 min full axon reconstruction in EM, 16.0 min control point placement in LM and EM). While this reduction in matching effort is well applicable to settings in which axonal varicosities can be detected, the more general but more laborious FluoEM approach of locally dense axonal reconstruction is expected to be applicable to other settings (for example transit axons that rarely form synapses). Together, we conclude that FluoEM is effective and efficient for LM-to-EM registration.

## Comparison of axon and synapse detection in LM and EM

The axons matched between the LM and EM volumes provided a dataset to calibrate in hindsight, how well axonal trajectories and branches can be reconstructed and how faithfully synapses can be detected in fluorescence LM data (*Figure 6*).

We first asked a different expert annotator to reconstruct the previously matched axons in the LM dataset, this time without an explicit bias to stop reconstruction at uncertain locations (cf. *Figure 2c*), and without knowledge of the matched EM axon's shape or trajectory. We then compared these axonal reconstructions performed in the LM dataset with the corresponding EM trajectories of the axons, which we considered the ground truth for the detection of branchpoints, the determination of the axonal morphology and the presence of presynaptic boutons (*Figure 6a–f*). We found that while a large fraction of axons were reconstructed faithfully at the LM level (27 of 38 axons, 71%), in 29% of cases the LM reconstruction contained an error (*Figure 6f*). The largest fraction of errors was related to incorrectly terminated reconstructions ('stop', 38%, *Figure 6b,f*), followed by an incorrect continuation of the LM-reconstructed axon at locations where multiple fluorescent axons overlapped (31% of errors, *Figure 6c,f*). Furthermore, missed (23%) and added (8%) branches occurred (*Figure 6d,e,f*).

Together, this data illustrates that at the chosen labeling density, fluorescence-based axon reconstruction in cortex is error-prone. Labeling density was 1.2 ± 0.2% (mean ± s.d., n = 115 planes) for the tdTomato channel and 1.4 ± 0.4% (mean ± s.d., n = 115 planes) for the eGFP channel; error rates evaluated separately for axons in each channel were 30% of axons (tdTomato) and 27% of axons (eGFP), respectively.

We then identified 255 axonal varicosities in the LM data and investigated their ultrastructure at the EM level based on the successfully matched axonal counterparts (*Figure 6a,g–o*). As the examples indicate (*Figure 6g–m*), many varicosities in the LM in fact correspond to presynaptic boutons (63%, n = 161 of 255, *Figure 6h,k,l*). A substantial fraction of LM-varicosities, however, are without evidence for a presynaptic specialization and instead contain only mitochondria (n = 54, *Figure 6g,j*), or are hollow thickenings (n = 39, *Figure 6i*) or myelinated (n = 1, *Figure 6m*). Notably, while the inference of synaptic boutons from LM varicosities was dependent on the size of the LM

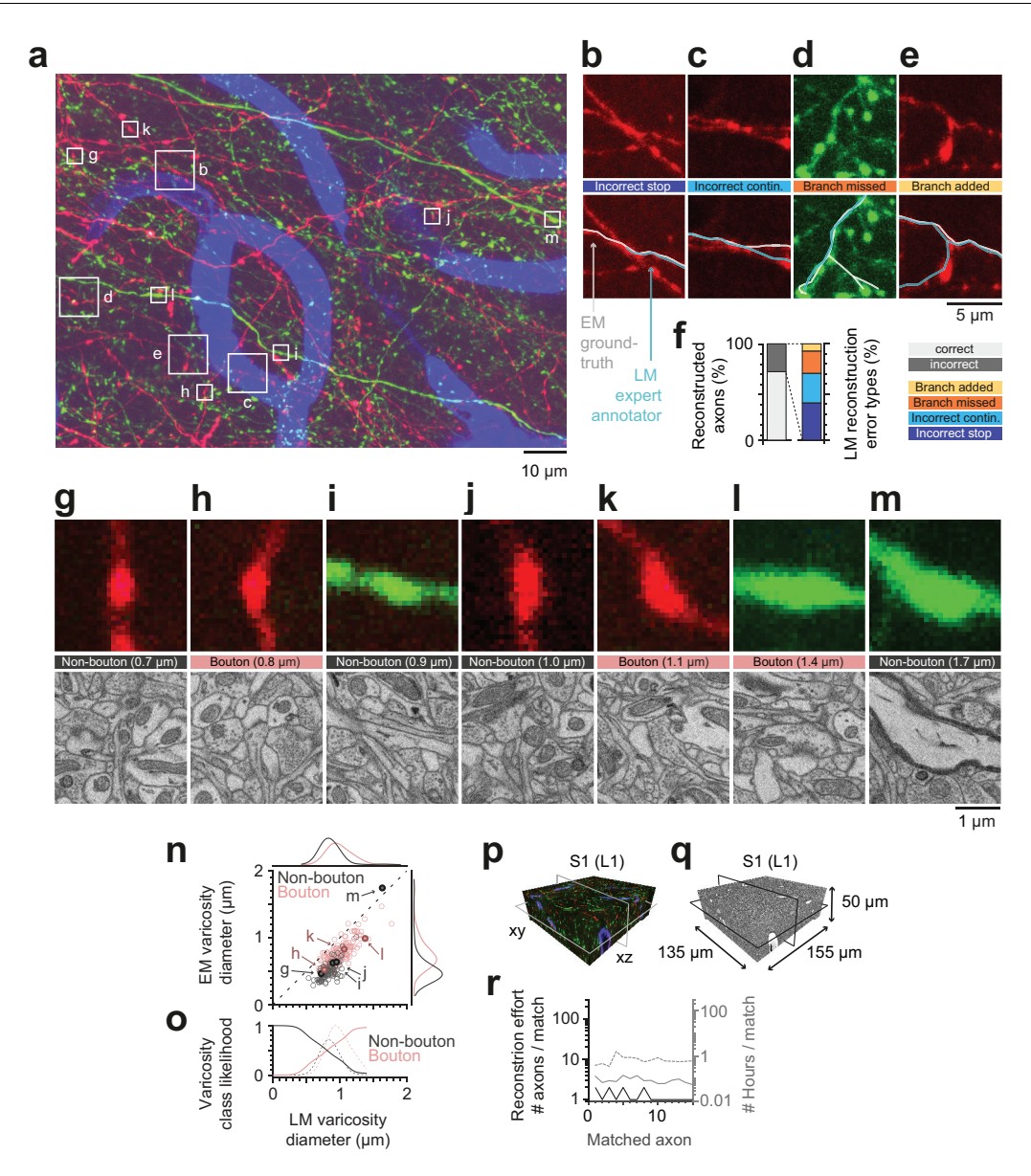

**Figure 6.** Calibration of axons and synapses in 3D fluorescent data using FluoEM, and reproduction experiment. (**a**) Examples of axonal trajectories and axonal varicosities in 3D LM data (Maximum-intensity projection from 42 image planes at 17–35 μm depth; dataset and color code as in *Figure 3*). (**b–f**) Post-hoc comparison of axonal reconstructions in LM data and their EM counterparts as identified by FluoEM, which were considered ground truth. Examples of LM reconstruction errors (**b–e**) and their prevalence (**f**). Note that while about 70% of axons were correctly reconstructed at the LM level, for about 30%, reconstruction errors such as missed branches or incorrect continuations occurred. Direct links to example locations in the dataset for viewing in webKnossos: b, https://wklink.org/1500 and https://wklink.org/8693; c, https://wklink.org/2441 and https://wklink.org/8447; d, https://wklink.org/5600 and https://wklink.org/4519; e, https://wklink.org/1977 and https://wklink.org/6578. (**g–m**) Validation of presynaptic bouton detection in fluorescence data: Examples of axonal varicosities imaged using LM (top row) and EM (bottom row). EM was used to determine synaptic (bouton) vs. non-synaptic varicosities. Size of varicosity reported as sphere-equivalent diameter (see Materials and methods). Direct links to example locations in the dataset for viewing in webKnossos: g, https://wklink.org/3721 and https://wklink.org/8170; h, https://wklink.org/6118 and https://wklink.org/7611; i, https://wklink.org/7447 and https://wklink.org/3702; j, https://wklink.org/9694 and https://wklink.org/6772; k, https://wklink.org/4867 and https://wklink.org/0276; l, https://wklink.org/7809 and https://wklink.org/2224; m, https://wklink.org/7269 and https://wklink.org/5832. (**n**) Relation between the sphere equivalent diameter of axonal varicosities imaged in EM vs. LM for synaptic (red, bouton) and non-synaptic (black) varicosities. (**o**) Likelihood of varicosities to be synaptic (red) vs. non-synaptic (black) over LM varicosity sphere equivalent diameter, computed from the respective (prevalence-weighted, kernel density estimated) probability densities (dashed, see Materials and methods). (**p–r**) Second FluoEM experiment from upper cortical L1 in S1 cortex (p: LM dataset subvolume corresponding to (**q**); q: EM high-res dataset with dimensions) and (**r**) reconstruction effort for LM-to-EM matching of axons (labels as in *Figure 5e*).

*Figure 6 continued on next page*

*Figure 6 continued*

DOI: https://doi.org/10.7554/eLife.38976.021

The following source data is available for figure 6:

**Source data 1.** Relation between sphere equivalent diameter in LM and EM for synaptic and non-synaptic axonal varicosities (*Figure 6n*).
DOI: https://doi.org/10.7554/eLife.38976.022
**Source data 2.** Sphere diameter dependent synaptic and non-synaptic varicosity likelihood (*Figure 6o*).
DOI: https://doi.org/10.7554/eLife.38976.023
**Source data 3.** Reconstruction effort reproduction experiment (*Figure 6r*).
DOI: https://doi.org/10.7554/eLife.38976.024

varicosity (*Figure 6n*) with larger varicosities more likely synaptic, there was no LM varicosity size range that allowed unequivocal inference of a synapse from the LM morphology (*Figure 6n,o*).

## Reproduction experiment

Finally, in order to assure the reproducibility of FluoEM, we conducted a second correlated imaging experiment in layer 1 of mouse S1 cortex (LM dataset sized $(1027 \times 820 \times 84)$ $\mu m^3$, EM dataset sized $(155 \times 135 \times 50)$ $\mu m^3$, *Figure 6p–q*), in which we successfully matched 15 axons between LM and EM as a proof of principle. The total time investment (*Figure 6r*) for the second matching experiment was 10 hr, which included LM-to-EM axon matching (1.3 hr), full EM-based axon reconstruction (2.4 hr) and LM-EM control point placement (6.5 hr).

## Discussion

We report FluoEM, a set of experimental and computational tools for the virtual labeling of multiple axonal projection sources in connectomic 3D EM data of mammalian nervous tissue. At its core is the notion that dense EM circuit reconstruction allows to reformulate the alignment between data obtained at micrometer resolution (LM) to data obtained at nanometer resolution (EM) to the best-matching between a sparse subset of labeled axons (LM) and all possible axons (EM). We find that in mammalian neocortex, densely reconstructed axons have unique trajectories on a spatial scale that is compatible with currently achievable 3D EM dataset sizes, allowing the FluoEM concept to be successfully applied. With this, FluoEM allows the enriching of EM-based connectomes in mammalian neuropil with the large range of multi-color encoded information already successfully expressed in axons at the light-microscopic level.

One key prerequisite for FluoEM is the uniqueness of axonal trajectories in dense neuropil. We have so far determined the uniqueness length for several layers of the mouse cerebral cortex (*Figure 2*), where this length is consistently below 40 µm (90th percentile, see *Figure 2*), thus in a range that can be faithfully reconstructed at the LM level (*Figure 2a–c*). For a broad range of cortical and subcortical nervous tissue (basal ganglia, thalamus, hypothalamus, nuclei of the brain stem etc.) in which locally dense circuits coexist with incoming projection axons from distant sources, FluoEM can be applied on the spatial scales described here. Cases in which dense neuropil contains highly aniso-tropic axons (for example axon bundles in the white matter of mammalian brains) will likely imply a longer scale on which axonal trajectories become unique, and therefore require longer reconstruction lengths in the fluorescence data.

We have developed FluoEM using a particular 3D EM imaging technique, SBEM (*Denk and Horstmann, 2004*). Since FluoEM only relies on the underlying geometry of axonal trajectories, which we have shown to be unique on scales achievable with all current 3D EM techniques (*Figure 2g–k*), FluoEM is not restricted to the combination with SBEM. Rather, other 3D EM imaging approaches, especially FIB-SEM (*Heymann et al., 2006*; *Knott et al., 2008*; *Hayworth et al., 2015*; *Xu et al., 2017*) and ATUM-SEM (*Schalek et al., 2011*; *Kasthuri et al., 2015*) that provide faithful axonal reconstructions over at least about 40–50 µm extent in three dimensions will be useable for the presented approach.

While FluoEM in principle allows the identification of as many axonal projection sources in a single connectomic experiment as can be encoded at the light-microscopic level, it is practically limited by the number of fluorescence channels that can be acquired in sequence without exerting damage to

the tissue. In our experiments, tissue exposed to 2 hr of continuous LM imaging with acquisition of three parallel fluorescence channels was still well usable for subsequent 3D EM circuit reconstruction. To acquire more fluorescence channels without increasing the local photon dose in the tissue, one can exploit the fact that in FluoEM, it is not necessary to image the axons of interest in their entirety at the LM level. Rather, it is sufficient to LM-image volumes of 40–50 μm in extent, as long as all axons of interest pass through that volume, and to reconstruct the complete axons in EM. Therefore, a setting in which the volume of interest is split into several subvolumes, in each of which a limited number of fluorescence channels is acquired, could multiplex the color range of FluoEM (*Figure 7a–b*) without increasing the exposure time in the LM step.

In LM imaging of mammalian nervous tissue, labeling density of axons poses a notorious limitation as long as axonal trajectories were to be reconstructed from the LM data (labeling more than about 1 in 1000 to 10,000 axons impedes reconstruction (*Helmstaedter, 2013*); and LM reconstruction errors may increase substantially with increased labeling density, see *Figure 6b–f*). With FluoEM, labeling density is only limited by the need to follow axons over the uniqueness length (*Figures 2* and *3*). Since this depends on the overall alignment precision (*Figure 2k*), a setting in which one fluorescence channel contains sparsely labeled axons to provide an accurate pre-alignment (see *Figure 5g,h*), and the other fluorescence channels are densely labeled is a possible solution, in which LM-reconstructable stretches of axons as short as 5 μm would be sufficient for co-registration (*Figures 2k* and *5h*). For the experiment described here, we titrated the labeling density by adjusting the injection volume while keeping the virus titer constant (see Materials and methods). Depending on the distance and connectivity of the projection sources to the FluoEM target area this parameter might need to be adjusted. An extreme example of sparse labeling could be the parallel intracellular injection of two or more (neighboring) neurons in one part of cortex (*Figure 7c*) and the investigation of their convergent projections in other parts of the brain (*Figure 7c*); in this case, FluoEM could be used to study the synaptic output logic of multiple identified neurons at single-cell and single-synapse level.

Finally, the reconstruction work load in FluoEM requires a quantitative discussion. In settings where axonal varicosities can be used as additional constraints on the matching (*Figure 5a–d*), the LM-to-EM match consumed about 20–30 min per axon (including the full EM reconstruction of the axon and the placement of control points). With this, a full matching of a sparsely stained volume

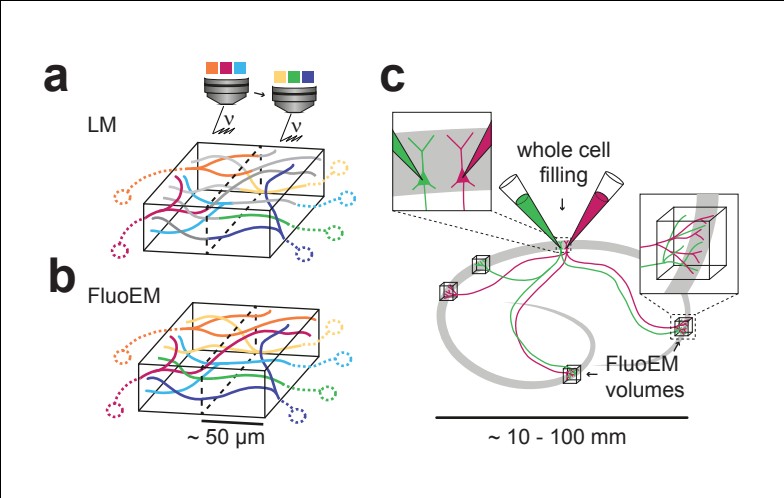

**Figure 7.** Possible extensions of FluoEM (**a,b**) Color multiplexing strategy to increase color space without increasing photon dose in FluoEM. If labeled axons traverse a large volume, then 3D LM imaging can be split into subvolumes of minimal dimension about 50 μm (given by $d_{recon}$ and $d_{unique}$, see *Figure 2*) (**a**), and axon identity can be propagated into the remaining volume via EM reconstruction (**b**) using the FluoEM approach. (**c**) Instead of identifying input projections in dense 3D EM by fluorescence labeling, the output connectivity of (multiple) fluorescently labeled neurons in convergent target regions could be studied using FluoEM (see sketch).
DOI: https://doi.org/10.7554/eLife.38976.025

sized $(1 \times 1 \times 0.1)$ mm$^3$, for example a large area of layer one in the mouse neocortex, would thus likely consume about 5300 work hours in FluoEM (using webKnossos (*Boergens et al., 2017*), see Materials and methods for estimates). While this is a considerable time investment, it favorably compares to other resource investments in biomedical imaging. Furthermore, with the development of increasingly reliable automated approaches for the reconstruction of axons and synapses at the EM level (*Berning et al., 2015*; *Dorkenwald et al., 2017*; *Staffler et al., 2017*; *Januszewski et al., 2018*), a further speedup of FluoEM is conceivable in which partially automated matching contributes. In mammalian circuits where axons may not show local diameter variations that can be used for co-registration, the FluoEM approach using dense axonal reconstruction (*Figure 2*) is applicable.

Together, the presented methods overcome the major hurdle of limited color space for identifying multiple axonal input sources in parallel in large-scale and dense electron microscopic reconstruction of mammalian neuronal circuits.

# Materials and methods

## Key resources table

| Reagent type | Designation | Source or reference | Identifiers | Additional information |
|---|---|---|---|---|
| Strain, strain background (*Adeno-associated virus*) | AAV1.CAG.FLEX.EGFP.WPRE.bGH | Penn Vector core | AllenInstitute854 | Lot: V3807TI-R-DL |
| Strain, strain background (*Adeno-associated virus*) | AAV1.CAG.FLEX.tdTomato.WPRE.bGH | Penn Vector core | AllenInstitute864 | Lot: V3675TI-Pool |
| Strain, strain background (*Adeno-associated virus*) | AAV1.CamKII0.4.Cre.SV40 | Penn Vector core | | Lot: CS0302 |
| Chemical compound, drug | 1,1'-Dioctadecyl-3,3,3',3'-Tetramethylindodicarbocyanine | Invitrogen/Thermo Fisher | D7757 | |
| Chemical compound, drug | Paraformaldehyde | Sigma Aldrich/Merck | P6148 | |
| Chemical compound, drug | Glutaraldehyde | Serva | 23116 | |
| Chemical compound, drug | Sodium Cacodylate | Serva | 15540 | |
| Chemical compound, drug | Osmium Tetroxide | Serva | 31253 | |
| Chemical compound, drug | Ferrocyanide (Potassium hexacyanoferrate trihydrate) | Sigma Aldrich/Merck | 60279 | |
| Chemical compound, drug | Thiocarbohydrazide | Sigma Aldrich/Merck | 223220 | |
| Chemical compound, drug | Uranyl acetate | Serva | 77870 | |
| Chemical compound, drug | Lead (II) nitrate | Sigma Aldrich/Merck | 467790 | |
| Chemical compound, drug | L-Aspartic acid | Serva | 14180 | |
| Commercial assay or kit | Spurr low viscosity embedding kit | Sigma Aldrich/Merck | EM0300 | |

## Animal experiments

All experimental procedures were performed according to the law of animal experimentation issued by the German Federal Government under the supervision of local ethics committees and according to the guidelines of the Max Planck Society. The experimental procedures were approved by Regierungspräsidium Darmstadt, V54 - 19c20/15 F126/1015.

## Virus injection

Young adult (p28) male wild-type mice (C57BL/6 j) were anesthetized with isoflurane (induction: 4%, maintenance: 2%) in medical oxygen and fixed in a stereotaxic frame (Model 1900, Kopf Instruments, USA). The core body temperature was maintained at 37°C using a feedback-controlled heating pad (DC Temperature Control System, FHC, USA). Systemic analgesia was provided by subcutaneous injection of 2 mg/kg Meloxicam (Metacam, Boeringer-Ingelheim, Germany) and 100 mg/kg Metamizol (Metamizol WDT, WDT, Germany) prior to surgery. Local anesthesia was provided by injecting 16.7 mg/kg Ropivacaine (Naropin, AstraZeneca, Switzerland) under the scalp. To anterogradely label projections originating from M1 and S2 cortex, adeno-associated viruses expressing eGFP (AAV1. CAG.FLEX.EGFP, $2.4 \times 10^{13}$ GC/ml, Penn Vector Core, USA) and tdTomato (AAV1.CAG.FLEX.tdTomato, $1.5 \times 10^{13}$ GC/ml, Penn Vector Core, USA) were injected using freshly pulled 1.5 mm borosilicate glass capillaries at the following coordinates: 1.0 mm anterior of bregma, 1.0 mm lateral of the midline, 0.7 mm below the cortical surface (M1) and 0.6 mm posterior of bregma, 4.1 mm lateral of the midline, 1.3 mm below the cortical surface (S2). Both AAV.FP solutions were mixed in a 2:1 ratio with AAV.Cre solution (AAV1.CamKII0.4.Cre, $4.8 \times 10^{13}$ GC/ml, Penn Vector Core, USA) prior to injection. At each site approximately 50 nl were injected using a pressure injection system (PDES, NPI, Germany).

## Transcardial perfusion

Three weeks after virus injection, mice were subcutaneously injected with 0.1 mg/kg Buprenorphine (Buprenovet, Recipharm, France) and 100 mg/kg Metamizol. General aneasthesia was introduced with 4% isoflurane and maintained with 3% isoflurane in medical oxygen. Mice were transcardially perfused with 10–15 ml flushing solution containing a lipophilic far-red fluorescent dye for blood vessel labeling (150 mM Sodium Cacodylate, 5 µM 1,1'-Dioctadecyl-3,3,3',3'-Tetramethylindodicarbocyanine (DiD, Thermo Fisher, USA), pH 7.4) followed by 50–60 ml fixative solution (80 mM Sodium Cacodylate, 2.5% Paraformaldehyde, 1.25% Glutaraldehyde, 0.5% $CaCl_2$, pH 7.4) using a syringe pump (PHD Ultra, Harvard Apparatus, USA) at a flow rate of 10 ml / min. After perfusion, mice were decapitated and the connective tissue around the skull was removed. The head was immersed in fixative solution for approximately 24 hr at 4°C.

## Sampling

After extracting the brain from the skull and placing it in a dish filled with storage buffer (150 mM Sodium Cacodylate, pH 7.4) coronal vibratome (HM650V, Thermo Fisher Scientific, USA) sections of 100–150 µm thickness were made until the target position along the rostro-caudal axis was reached at 1.6 mm posterior of bregma. A coronal slab of 1 mm thickness was then cut off the remaining brain from which a cylindrical sample containing cortical layers 1–5 was extracted from 1.3 mm lateral of the midline using a medical 1.5 mm biopsy punch (Integra, Miltex, USA).

## Confocal imaging

Prior to confocal imaging, the fixed brain tissue sample was washed for 30 min in storage buffer containing a mild reducing agent (150 mM Sodium Cacodylate, 2 mM reduced Glutathion, pH 7.4) and then transferred into a storage buffer-filled imaging chamber (Cover Well, Grace Bio-Labs, USA). Using an inverted Confocal Laser Scanning Microscope (LSM 880, Zeiss, Germany) and a 40x water objective (C-Apochromat 40x/1.2 W Korr FCS M27, Zeiss, Germany), a three-channel (eGFP labeled axons, tdTomato labeled axons, DiD labeled blood vessels) image stack of dimensions ($828 \times 828 \times 75$) µm$^3$ was acquired at a voxel size of ($104 \times 104 \times 444$) nm$^3$ in a single scanning track using 488 nm and 561 nm laser lines.

## EM staining

Immediately after imaging the sample was removed from the imaging chamber and immersed in 2% $OsO_4$ aqueous solution (Serva, Germany) in cacodylate buffer (150 mM, pH 7.4). After 90 min the solution was replaced by 2.5% ferrocyanide (Sigma-Aldrich, USA) in cacodylate buffer (150 mM, pH 7.4) and incubated for another 90 min. The solution was subsequently exchanged for 2% $OsO_4$ in cacodylate buffer (150 mM, ph 7.4) and incubated for another 45 min. The sample was then washed first for 30 min in cacodylate buffer and then for another 30 min in ultrapure water. Next, the sample

was incubated in 1% thiocarbohydrazide (Sigma-Aldrich, USA) at 40°C for 45 min and washed twice in ultrapure water for 30 min each before immersing it in 2% unbuffered $OsO_4$ aqueous solution. After 90 min the sample was again washed twice for 30 min each in ultrapure water and subsequently immersed in 1% uranyl acetate (Serva, Germany) aqueous solution at 4°C overnight. On the next day, the sample (still immersed in uranyl acetate solution) was heated to 50°C in an oven (Universal Oven Um, Memmert, Germany) for 120 min. After two washing steps with ultrapure water lasting 30 min each, the sample was incubated in a 20 mM lead aspartate (Sigma-Aldrich, USA) solution (adjusted to pH 5 with 1N KOH) at 50°C for 120 min. The sample was then washed twice in ultrapure water again for 30 min each. Unless specified otherwise all steps were carried out at room temperature.

## EM embedding

After concluding the staining procedure the sample was dehydrated by immersing it for 30 min each at 4°C in a graded ethanol (Serva, Germany) series of 50% ethanol in ultrapure water, 75% ethanol in ultrapure water and 100% ethanol, respectively. The sample was then immersed three times in acetone (Serva, Germany) 30 min each at room temperature. After completing the dehydration steps, the sample was infiltrated with a 1:1 mixture of acetone and Spurr's resin (Sigma Aldrich, USA) with a component ratio of 4.1 g ERL 4221, 0.95 g DER 736 and 5.9 g NSA and 1% DMAE at room temperature for 12 hr on a 45 degree rotator in open 2 ml reaction tubes (Eppendorf, Germany). Infiltrated samples were then incubated in pure resin for 6 hr, placed in embedding molds and cured in a preheated oven at 70°C for 48–72 hr.

## EM stack acquisition

The embedded sample was trimmed with a diamond-head milling machine (EM Trim, Leica, Germany) to a cube of approximately $(1.5 \times 1 \times 1)$ mm$^3$ dimension and mounted on an aluminium stub with epoxy glue (Uhu Plus Schnellfest, Uhu, Germany) in tangential orientation, so that the pia mater was located at the top of the block face. The thin layer of resin covering pia was carefully removed with a diamond-knife ultramicrotome (UC7, Leica, Germany). The smoothed tissue block was then gold-coated (ACE600, Leica, Germany) and placed in a custom-built SBEM microtome (courtesy of W. Denk) mounted inside the chamber of a scanning electron microscope (FEI Verios, Thermo Fisher Scientific, USA). Microtome movements and image acquisition were controlled using custom-written software packages interfacing with motor controllers and the microscope software's API. A low-resolution image stack of dimensions $(1500 \times 1000 \times 10)$ μm$^3$ was acquired at a pixel size of $(270 \times 270)$ nm$^2$ and a cutting thickness of 200 nm. After correspondences with the confocal image stack were successfully identified, a high resolution image stack sized $(130 \times 110 \times 85)$ μm$^3$ was acquired at a pixel size of $(11.24 \times 11.24)$ nm$^2$ and a cutting thickness of 30 nm (*Figure 3*).

## Image post-processing

The LM image stack (*Figure 2a–c*, *Figure 3*) consisted of $16 \times 153$ (xy x z) image tiles ($2048 \times 2048$ pixels each) acquired at eight bit color depth for each of the three color channels, resulting in 29 GB of uncompressed data. The 2448 image tiles were stitched into a global reference frame using the Zen software package (Zeiss, Germany). The LM image volume was then rotated so that the orientation of the superficial blood vessel pattern coarsely matched the orientation of the blood vessel pattern visible in the EM overview images. The images were then converted into the Knossos (*Helmstaedter et al., 2011*) format and transferred to a data-store hosted at the Max Planck Compute Center in Garching from which they can be accessed via our online data viewer webKnossos (*Boergens et al., 2017*, webknossos.org).

The EM image stack acquired was comprised of $20 \times 2768$ (xy x z) image tiles ($3072 \times 2048$ pixels each) with eight bit color depth, resulting in 326 GB of uncompressed data. The 55360 resulting tiles were stitched into a global reference frame using custom written Matlab software based on the approach described in *Briggman et al. (2011)*. However, instead of cross-correlation we used SURF feature detection to estimate relative shift vectors and their confidence. We then used a weighted least-squares approach to retrieve a robust globally optimal solution given these parameters. The EM data was then also converted into the Knossos format and transferred to a data-store hosted at

the Max Planck Compute Center in Garching from which they can be accessed via our online data viewer webKnossos.

## Measurement of LM axon reconstructable pathlength (d$_{recon}$)

For estimating the distribution of the reconstructable pathlength $d_{recon}$3 in the LM data, a (40 x 40 x 48) μm$^3$ bounding box was placed in a subvolume of the dataset with representative labeling density and all visible axons traversing this bounding box were skeleton-reconstructed independently by two expert annotators using webKnossos. Then the path-lengths of the resulting skeleton representations were measured by calculating the cumulative sum of the Euclidian distances between all connected nodes and a histogram over the distances was computed (*Figure 2c*).

## Measurement of EM axon uniqueness pathlength (d$_{unique}$)

To measure the uniqueness distance $d_{unique}$ of cortical axons in different cortical depths, a rectangular bounding box of edge length $\lambda_{align}$ was placed at the center of each dataset and all axons traversing this bounding box were reconstructed, resulting in a set of axon skeletons. To measure $d_{unique}$ for a given axon $ax_i$ a marching sphere with radius $r \approx \lambda_{align}$ was placed iteratively around each node $n_i^k$ and all neighboring axons $ax_j$ with at least one of their nodes $n_j^l$ residing within the sphere so that $\left\| n_i^k - n_j^l \right\| \leq r$ were identified, resulting in a binary neighbor vector indicating the presence (*true*) or absence (*false*) of each putative neighboring axon at each sphere (node) location. The results were then sorted according to the Euclidian distance between the respective sphere (node) location and the bounding box origin. Whenever a value of the distance sorted neighbor vector changed from true to false at consecutive distances $d$ and $d + 1$ the respective distance $d$ was considered the dropout distance for a given neighbor axon $ax_j$. We define $d_{unique}$ and $d_{unique}^{90}$ as the Euclidean distances from the origin for which all or 90% neighboring axons have dropped out, respectively (*Figure 2f,g–i*).

## Coarse LM-EM alignment

Based on the characteristic pattern of large superficial blood vessels visible both in the LM data and the low-resolution EM data (*Figure 3d,e*) a coarse correspondence was visually established between the two datasets. An approximate bounding box was applied to the LM dataset defining the subvolume corresponding to the acquired high resolution EM dataset. Subsequently, all blood vessels were traced using webKnossos. Based on characteristic blood vessel bifurcations, seven pairs of blood vessel skeleton nodes were defined as corresponding control point pairs $CP = (CP_{EM}, CP_{LM})$ with $CP_{EM} = \{cp_{EM}^1, cp_{EM}^2, \ldots, cp_{EM}^7\}$ and $CP_{LM} = \{cp_{LM}^1, cp_{LM}^2, \ldots, cp_{LM}^7\}$ used to constrain a three-dimensional affine transformation $AT_{BV}$ (*Figure 3h,o*).

## Affine registration

Given a set of control point pairs and an initial scaling vector resulting from the ratio between the nominal LM and EM voxel sizes we computed an initial affine transformation using Horn's quaternion-based method (*Horn, 1987*). We then iteratively optimized the scaling vector by minimizing the mean squared residual registration error. We then applied the resulting transformation to the nodes of skeleton reconstructed LM blood vessels or axons to obtain affine transformed skeletons registered to their corresponding structures in EM reference space.

## Free-from registration

Given a set of control point pairs and an initial scaling vector we first computed the optimal affine registration as described above. We then used a b-spline interpolator based approach (*Lee et al., 1997*; *Rueckert et al., 1999*) to compute an optimal free-form registration given a set of control points, an initial b-spline knot spacing and a number of bisecting mesh refinement steps. We found a combination of 32 μm initial spacing and four refinement steps to give us an optimal compromise between registration precision and robustness. We then applied the resulting deformation grid to the nodes of skeleton reconstructed LM axons to obtain free-form transformed skeletons registered to their corresponding structures in EM reference space.

### Initial axon matching

As a candidate for the initial axon matching an axon $ax1_{LM}$ in spatial vicinity of a blood vessel bifurcation was picked and reconstructed in the LM dataset using webKnossos. The previously constrained affine transformation was then used to transform $ax1_{LM}$ into the EM reference space so that $AT_{BV}\left(ax1_{LM}\right) \rightarrow ax1_{LM}^{AT}$. Because $ax1_{LM}$ was chosen close to one of the corresponding blood vessel bifurcations, it was possible to estimate the local affine transformation error by applying the same transformation to the blood vessel skeletons and measuring the offset at the closest bifurcation. Based on this offset, a local translative correction $AT_{trans}^{BV}$ was defined and used to locally correct the offset of the first axon candidate: $AT_{trans}^{BV}\left(ax1_{LM}^{AT}\right) \rightarrow ax1_{LM}^{AT'}$. A bounding box was then defined around a node $n \in N\left(ax1_{LM}^{AT'}\right)$ close to the blood vessel bifurcation and all axons traversing this bounding box were reconstructed. The true corresponding axon $ax1_{EM}$ amongst all of these possible axon candidates was selected by the marching sphere method (see Materials and methods: measurement of EM axon uniqueness pathlength, *Figure 2*, *Figure 4i*) followed by a validation based on varicosity patterns (*Figure 4j*).

### Iterative axon matching and free-form transformation

After successfully matching a given axon we used its varicosities and branchpoints as additional constraints to update our registration, thereby increasing local registration precision substantially. By applying these updates and carefully choosing an axon in the spatial vicinity of a previously matched as the next matching candidate, we were able to reduce the search volume from (5 µm)$^3$ to (2–3 µm)$^3$ depending on the distance of the seed point on the candidate axon to the closest previously matched constraint and the general registration precision in that area. Furthermore, by using a varicosity as a seed point around which the search volume would be centered we were able to even further reduce the number potential EM candidates to only such axons displaying such a structural feature within the reduced search volume (*Figure 5a–d*). Only if we did not succeed in finding the match within the small bounding box, its volume was incrementally increased until the match was found. Using this method, we were able to identify the match for every axon we attempted.

### Time investment for axon matching

To assess the work hour investment when using FluoEM we recorded the total time required to find a given LM-to-EM match using the internal time tracking in webKnossos. The work load for full EM-based axon reconstruction was estimated using the average tracing speed in orthogonal webKnossos tracing mode (*Boergens et al., 2017*) multiplied by the respective axon path length. For the control point placement, we estimated an average time investment of 2 min per control point. In our experience, this is a conservative estimate, since control point placement can be substantially accelerated when using transformed axon skeletons containing for example varicosity or branchpoint annotations from one modality as a guiding template for the other modality. Control point management and the handling of corresponding axon skeletons was carried out using the SkelReg Matlab class, which is an important part of the FluoEM software package provided here (see https://gitlab.mpcdf.mpg.de/connectomics/FluoEM; copy archived at https://github.com/elifesciences-publications/FluoEM).

### Registration error measurements

The residual registration error of the blood vessel-based affine registration was measured by applying the affine transformation $AT^{BV}$ to the nodes of the LM blood vessel skeleton representations and measuring the Euclidean distances between the EM control points $CP_{EM}$ and the transformed LM control points $CP_{LM}^{AT}$ located on the respective skeletons. In this case we used the same $n(CP_{residual}) = 7$ control point pairs to constrain the transformation and measure the residuals (*Figure 3o*).

The residual registration error of the axon-based affine registration was measured by applying the affine transformation $AT$ to the nodes of the LM axon skeleton representations and measuring the Euclidean distances between the EM control points $CP_{EM}$ and the transformed LM control points $CP_{LM}^{AT}$ located on the respective skeletons. We randomly picked $n(CP_{residual}) = 30$ control point pairs,

none of which had been used before to constrain the transformation, to measure the residuals (*Figure 5h*).

The residual registration error of the axon-based free-form registration was measured by successively applying the affine transformation $AT$ followed by the free-form transformation $FT$ to the nodes of the LM axon skeleton representations and measuring the Euclidean distances between the EM control points $CP_{EM}$ and the transformed LM control points $CP_{LM}^{FT}$ located on the respective skeletons. We randomly picked $n(CP_{residual}) = 30$ control point pairs, none of which had been used before to measure the residuals (*Figure 5h*).

To measure the effect the number of control points exerts on the residual registration error we randomly picked $n(CP_{constrain}) = [5, 10, 25, 50, 75, 100, 150, 200, 250]$ control points from a population of $n(CP_{total}) = 284$ control points and computed the registration. We then randomly picked an additional $n(CP_{residual}) = 30$ control points none of which were used in the step before for constraining the registration. We repeated this procedure 10 times for each number of control point pairs and reported the average and standard deviation of the respective sample residual mean (*Figure 5i*).

### Varicosity diameter measurements

We define axon varicosities as roughly ellipsoid-shaped local diameter increases for EM and as roughly ellipsoid-shaped local signal width and/or intensity increases for LM. The equivalent sphere diameters $d$ reported for the varicosities were calculated based on measured volumes $V$ using $d = 2\sqrt[3]{3V/4\pi}$. EM varicosity volumes $V_{EM}$ were measured by first computing an over-segmentation using SegEM (*Berning et al., 2015*), manually collecting segments belonging to a given varicosity and summing the number of voxels multiplied by the respective voxel volume. LM varicosity volumes were obtained by manually annotating the three principal axes of the ellipsoid-shaped varicosities, extracting the intensity profiles along these axes and fitting Gaussian functions to them. Finally, $V_{LM}$ was estimated as the ellipsoid volume resulting from the three full-width half-maximums of these Gaussians relative to the local baseline intensity (*Figure 6g–n*).

### Bouton and non-bouton likelihood estimation for varicosities

To estimate the likelihood of varicosities to be a synaptic bouton or not given its equivalent sphere diameters, we performed a kernel density estimation (bandwidth 100 nm) on the distribution of equivalent sphere diameters for the LM varicosities and weighted the estimated probability densities with the relative prevalence of the bouton vs non-bouton class (*Figure 6o*, dashed lines). The resulting bouton and non-bouton probabilities were then normalized to the sum of these probabilities to obtain the bouton and non-bouton likelihood (*Figure 6o*, solid lines), respectively.

### Comparison of axon detection in LM and EM

To test the reliability of axon reconstruction in LM we asked an expert annotator to again reconstruct all 38 axons that had been successfully matched before (by another expert), without knowledge of their shape or trajectories. To initialize these reconstructions, we randomly selected a central skeleton node and its two direct neighbors from each axon (only nodes of node degree two were considered). The expert annotator was then instructed to complete the LM axon reconstructions based on these initial locations using webKnossos; the expert was asked to reconstruct faithfully but also to attempt a full axonal reconstruction (thus relieving the bias towards split reconstructions applied in the calibrations of *Figure 2c*). The resulting reconstructions were then compared to the EM skeletons by visual inspection, and errors counted (*Figure 6a–f*).

### FluoEM for larger volumes: extrapolation

In a $(130 \times 110 \times 70)\ \mu m^3$ volume, we matched 38 axons constituting approximately 36% of the total fluorescence signal in both color channels. Assuming an average matching time effort of 5 min per axon, an average EM reconstruction time of 10 min per axon and an average control point placement time of 15 min per axon, a total time effort of 19 hr would be required to match the volume with 36% coverage. To achieve 100% coverage for the matched volume, 19 hr x (100% / 36%) ≈ 53 hr would be necessary. Thus, in order to match a volume of $(1000 \times 1000 \times 100)\ \mu m^3$ (comprising

e.g. a substantial patch of cortex L1) one can estimate the time effort to be approximately 53 hr x (1000 μm / 130 μm) x (1000 μm / 110 μm) x (100 μm / 70 μm) ≈ 5300 hr.

## Statistical tests

All statistical tests were performed using MATLAB.

To test whether the observed differences in $d_{unique}^{90}$ (*Figure 2j,k*) between L1, L2/3 and L4 are caused by the varying numbers of traversing axons encompassed in the $(5\ \mu m)^3$ bounding boxes (L1: n = 220 axons, L2/3: n = 207 axons, L4: n = 128 axons), we repeatedly (n = 30 draws) bootstrapped random subsamples of 128 axons from the L1 and L2/3 bounding box axon samples (without replacement) and measured $d_{unique}^{90}$. In both cases (L1: p = 0.54, L2/3: p = 0.14), the bootstrapped axon sets were indistinguishable from $d_{unique}^{90}$ of the L4 bounding box axon sample (one sample t-tests, see *Figure 2—figure supplement 1*).

To validate the initial axon match, we compared the similarity of the measured varicosity distributions on the LM axon and EM candidate axon (*Figure 4j*). To determine the probability of finding a matching varicosity pattern by chance, we performed a randomization test in which we repeatedly (n = 100,000 draws) assigned the measured number of EM varicosities (n = 11) to random positions (nodes) on the EM skeleton. We then calculated the mean Euclidian distance between the measured varicosity positions on the affine transformed LM axon and the (either measured or randomly drawn) positions on the EM axon and found it is unlikely ($p = 10^{-5}$) to find a match as good or better as the match based on the actually measured varicosity positions (*Figure 4—figure supplement 1*).

## Data and software availability

The fluorescence dataset and the 3D EM high-resolution and low-resolution datasets from L1 (*Figures 2–6*) are publicly available at demo.webknossos.org:

Low resolution EM dataset: https://demo.webknossos.org/datasets/FluoEM_2016-05-23_FD0144-2_st001_v1/view

High resolution EM dataset: https://demo.webknossos.org/datasets/FluoEM_2016-05-26_FD0144-2_v2s2s/view

Full field of view LM dataset: https://demo.webknossos.org/datasets/FluoEM_2016-06-02-FD0144_2_Confocal/view

Alternatively, direct short-links to the 3D datasets are provided in the figure legends and can be used to directly jump to the position and field of view as displayed in the respective figure panel.

The raw data for the reproduction experiment (*Figure 6p–r*) is available upon reasonable request.

All software routines are publicly available under the MIT license at fluoEM.brain.mpg.de and https://gitlab.mpcdf.mpg.de/connectomics/FluoEM (copy archived at https://github.com/elifes-ciences-publications/FluoEM).

## Acknowledgements

We thank Kevin Briggman and Gilles Laurent for discussions, Marcel Beining and Kun Song for comments on the manuscript, Alessandro Motta, Manuel Berning, Benedikt Staffler and Emmanuel Klinger for computational advice, Stephan Junek for support with light microscopic imaging, Johannes Letzkus for advice and support with viral injections, Heiko Wissler and Dalila Rustemovic for tracer management. We thank Matt Jacobson for providing the 'Absolute Orientation – Horn's method' Matlab central package we used in our affine registration workflow. We thank Dirk-Jan Kroon for providing the 'B-spline Grid, image and point registration' Matlab central package we used for our free-form registration workflow. We thank Susanne Babl, Lisa Bezzenberger, Alexander Brandt, Raphael Jakoby, Raphael Kneißl and Marc Kronawitter for annotator training and task management. We thank Aymun Al-Shaboti, Mahmoud Aly, Carolin Arras, Nagihan Aydin, Susanne Babl, Danny Baltissen, Anne Bamberg, Feron Basoeki, Natalie Berghaus, Alfred Berghoff, Lisa Bezzenberger, Nicola Böffinger, Svenja Bohne, Alexander Brandt, Oliver Brandt, Julius Buss, Lars Buxmann, Deniz Celik, Hanaa Charif, Nadia Cipta, Linda Decker, Kristina Desch, Tim Engelmann, Theresa Ernst, Jordi Espino Martinez, Benjamin Fani Sani, Theresa Fueller, Daniel Goffitzer, Victoria Gosch, Jennifer Hartel, Hanna Hees, Björn Heftrich, Julian Heller, Robert Hülse, Oana Ilea, Raphael

Jakob, Raphael Jakoby, Alexander Jost Lopez, Marcel Jüngling, Vanessa Kalbert, Mehmet Karabel, Lennart Kirchner, Raphael Kneißl, Plamen Kondev, Patricia König, Katharina Kramer, Franziska Krämer, Leonhard Kreppner, Marc Kronawitter, Jülide Kubat, Benjamin Kuhl, Derya Kurt, Eike Laube, Maria Leondaraki, Rebecca Lotz, Carina Lossnitzer, Luisa Lutz, Saskia Mehlmann, Isabell Metz, Nils Plath, Anna-Lena Possmayer, Leonard Präve, Maximilian Präve, Sylvia Reibeling, Sascha Reichel, Anna Rix, Claudio Sabatelli, Clemens Schumm, Lilli Schütz, Britta Stiehl, Kira Trares, Simon Umbach, Marco Werr, Jannik Winkelmeier, Timm Winkelmeier and Susanne Zimbelmann for neurite reconstruction.

## Additional information

### Competing interests

Moritz Helmstaedter: Reviewing editor, *eLife*. The other authors declare that no competing interests exist.

### Funding

| Funder | Grant reference number | Author |
| --- | --- | --- |
| Max-Planck-Gesellschaft | Open-access funding | Florian Drawitsch<br>Ali Karimi<br>Kevin M Boergens<br>Moritz Helmstaedter |

The funders had no role in study design, data collection and interpretation, or the decision to submit the work for publication.

### Author contributions

Florian Drawitsch, Resources, Data curation, Software, Formal analysis, Validation, Investigation, Visualization, Methodology, Writing—original draft; Ali Karimi, Data curation, Validation, Writing—review and editing; Kevin M Boergens, Data curation, Writing—review and editing; Moritz Helmstaedter, Conceptualization, Investigation, Visualization, Methodology, Writing—original draft, Project administration

### Author ORCIDs

Florian Drawitsch http://orcid.org/0000-0001-9543-1417
Moritz Helmstaedter http://orcid.org/0000-0001-7973-0767

### Ethics

Animal experimentation: All experimental procedures were performed according to the law of animal experimentation issued by the German Federal Government under the supervision of local ethics committees and according to the guidelines of the Max Planck Society. The experimental procedures were approved by Regierungspräsidium Darmstadt, V54 - 19c20/15 - F126/1015.

### Decision letter and Author response

Decision letter https://doi.org/10.7554/eLife.38976.034
Author response https://doi.org/10.7554/eLife.38976.035

## Additional files

### Supplementary files

• Transparent reporting form
DOI: https://doi.org/10.7554/eLife.38976.026

## Data availability

All imaging data is available for online browsing and annotation at demo.webknossos.org as detailed in the data availability section of the Methods.

The following datasets were generated:

| Author(s) | Year | Dataset title | Dataset URL | Database, license, and accessibility information |
|---|---|---|---|---|
| Drawitsch F, Helm-staedter M | 2018 | FluoEM low-res EM dataset | https://demo.webknos-sos.org/datasets/FluoEM_2016-05-23_FD0144-2_st001_v1/view | Openly accessible via webknossos.org (FluoEM_2016-05-23_FD0144-2_st001_v1) |
| Drawitsch F, Helm-staedter M | 2018 | FluoEM high-res EM dataset | https://demo.webknos-sos.org/datasets/FluoEM_2016-05-26_FD0144-2_v2s2s/view | Openly accessible via webknossos.org (FluoEM_2016-05-26_FD0144-2_v2s2s) |
| Drawitsch F, Helm-staedter M | 2018 | FluoEM LM dataset | https://demo.webknos-sos.org/datasets/FluoEM_2016-06-02-FD0144_2_Confocal/view | Openly accessible via webknossos.org (FluoEM_2016-06-02-FD0144_2_Confocal) |

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
