## [Decision Letter]

Thank you for submitting your article "FluoEM, Virtual labeling of axons in 3-dimensional electron microscopy data for long-range connectomics" for consideration by *eLife*. Your article has been reviewed by Eve Marder as the Senior Editor, a Reviewing Editor, and three reviewers. The following individual involved in review of your submission has agreed to reveal his identity: Harald F Hess (Reviewer #3).

The reviewers have discussed the reviews with one another and the Reviewing Editor has drafted this decision to help you prepare a revised submission.

Summary:

All of the reviewers were impressed with the importance and novelty of your work, and the high quality of the data. The full three reviews are included at the end of this letter, as there are a variety of specific and useful suggestions in them.

Reviewer #1:

This paper by Drawitsch et al., describes how fluorescently labeled axons imaged with light microscopy can be identified in EM image stacks using a process of computational matching of their trajectories. The paper is well presented, and the figures are appropriate.

This is an interesting paper and represents a small but significant step forward in the connectomics field. Identifying neurites belonging to different cells types in serial EM images has been achieved using a number of different methods over many years beginning with the work of Ed White in the 1970s. However, as the techniques for ultrastructural imaging have developed and vast volumes of brain tissue can now be visualized, tools for labeling different cell types in the EM are still very limited. Combining EM with LM is the obvious way forward but finding neurites in EM image stacks that were previously seen with LM is not straightforward. To date, this has been achieved with careful targeting of small regions using fiducial marks, and as the authors state, this is not relevant for connectomics' analysis. The current paper provides a detailed description of axon trajectories, giving some measurement of their uniqueness of length to allow a sorting process and selection of possible candidates. The final conclusion is that axon lengths of only less than 40 microns are needed for the method to work.

The authors provide all the datasets, which are exceptional, and it’s clear that very few laboratories could achieve imaging on such a scale. I am more enthusiastic about the results rather than the tools themselves as few groups have the resources for this approach.

I only have one question about the method. I fully appreciate that the authors describe a systematic computational approach for matching axons but while reading the paper and understanding how long this takes I wonder how quickly axons can be found by a human observer using the same datasets. Or is this even possible? In the methods part, the authors explain how they use the blood vessels to locate the region in the tissue from which the EM image stack was taken. Therefore, the region in which the axon can be found has been narrowed down considerably. If single axons are evident in the LM dataset, can these be found in the EM once it's clear where in the volume they should be located? As the authors describe how boutons are used to confirm the identity of the axons, could human observers achieve the same task more rapidly without having to make such large reconstructions of many axons?

*Reviewer #2:*

In this manuscript Drawitsch et al., present a workflow – FluoEM – for faithfully identifying correlated axon segments in sequentially imaged light microscopic and EM datasets. The approach is a significant advance in 3D connectomics as it allows labeled long-range axonal inputs to be mapped onto dense EM neuropil reconstructions. This approach has advantages over other correlative LM-EM approaches as it does not require any chemical label conversion and can potentially be used to study long-range axonal targets of genetically identifiable neurons from selected brain areas at nanometer resolutions afforded by EM. The results presented, especially the matching of varicosities to synapses is very satisfying and convincing. The experimental, computational methods described are sound and well documented. Together the approaches and tools described in this manuscript are likely to be impactful for the connectomics field and this paper merits publication in *eLife*.

1) In subsection “Comparison of axon and synapse detection in LM and EM” the authors compare axon and synapse detection in both LM and EM. While it is indisputable that the unambiguous assignment of synapses would require EM data the nearly 30% error rate in the assignment of axonal trajectories and branches in the LM data reported in this study appears unusually high. Is this likely due to high density of viral labeling. It would be useful to understand in more detail the effects of labeling sparsity in LM accuracy. Was the sparsity consistent across the 3 samples studied? Are the sparser regions of the neuropil associated with more accurate tracings?

2) The presentation of timing in various parts of the manuscript (Results section, Discussion section and the extrapolation in the Materials and methods section) is a bit confusing and could be better. For instance, what is the relative work load of completion needed for matching versus the time for reconstructing complete axons in the EM volume? It is also not clear if there are other bottleneck steps in scaling to larger volumes and if a linear extrapolation of axon segment matching times is relevant.

3) The current approach of correlative LM-EM is designed for looking at EM level connectivity of relatively dense axonal projections from different labeled 'source' brain areas. A complementary use case would be to look at input-output relationships of entire axonal arbors of single labeled neurons – the scales of such arbors could be much larger than the EM volume imaged in this study. Could the authors speculate on the feasibility of an approach wherein the acquisition of a high resolution EM dataset would be guided by the LM imaging of a very sparse labeled sample?

*Reviewer #3:*

The manuscript addresses a very important factor in extending the relevance of connectomics for mammalian brains, namely how to assign the source or destination of long-range projections associated with a target neuropil. Presently, the imaging and reconstruction of neural tissue volumes approaching 1 mm^3^ are becoming possible, yet a complete mammalian brain is out of range by 2-3 orders of magnitude. So, the identification of long-range input and output connections of a 1 mm^3^ region is clearly a major component to understanding the circuitry a reconstructed local region.

The research and manuscript are high quality and comprehensive. As just mentioned the motivation is well justified, the methodology for CLEM is well documented, as well as exemplary data sets and analysis software. The criteria for establishing positive or most likely neuron identification is described thoroughly in both a general level and in the fine detail of source code. The major concerns of error, e.g. error rates from light microscopy reconstruction are presented explicitly and shown how they can still be corrected to give very convincing correlative identification. Several different identification strategies e.g. path length, varicosities, etc. are compared as multiple cross confirming and extending options. Demonstrations in several locations in different layers of the cortex also prove the general applicability of this methodology.

One important parameter seems to be the axon labeling density. It seems that this must be controlled to get both as many as possible axons and yet sparse enough that they are separable in LM with the exemplary value of ~71% faithful reconstruction number. It would be helpful to have a rough number here and discussion of how that might be controlled or how that limits or expands the fraction of I/O projections into the EM volume.

There is without doubt a highly relevant, timely, and useful contribution to the field.

---

## [Author Response]

Reviewer #1:[…] The authors provide all the datasets, which are exceptional, and it’s clear that very few laboratories could achieve imaging on such a scale. I am more enthusiastic about the results rather than the tools themselves as few groups have the resources for this approach.

We thank the reviewer for the positive assessment. To further feature the data accessibility as lauded by the reviewer we inserted direct links to the 3D browser into the figure legends; with these the reader will be able to reproduce the figure panel by one click and start browsing the 3D data from there.

I only have one question about the method. I fully appreciate that the authors describe a systematic computational approach for matching axons but while reading the paper and understanding how long this takes I wonder how quickly axons can be found by a human observer using the same datasets. Or is this even possible? In the methods part, the authors explain how they use the blood vessels to locate the region in the tissue from which the EM image stack was taken. Therefore, the region in which the axon can be found has been narrowed down considerably. If single axons are evident in the LM dataset, can these be found in the EM once it's clear where in the volume they should be located? As the authors describe how boutons are used to confirm the identity of the axons, could human observers achieve the same task more rapidly without having to make such large reconstructions of many axons?

As soon as the axon of interest in LM can be narrowed down to a region of about 5 µm in extent (for example by running past a blood vessel bifurcation, or a cell body at this LM scale; this rough LM alignment distance we call λ _align_, e.g. Figure 3L-N, Figure 4E), the human observer will find the corresponding axon in EM; however in the general case, the human observer will have to reconstruct all 100-200 axons traversing this (5 µm)^3^ volume; then our data shows that the match will be unique for all cortical tissue investigated (Figure 2G-I). If in addition, the axon has varicosities visible in the LM (see Figure 5A-D), then this search will be significantly faster, down to only about 1-3 axons before a match can be found (Figure 5E and Figure 6R).

Reviewer #2:In this manuscript Drawitsch et al., present a workflow – FluoEM – for faithfully identifying correlated axon segments in sequentially imaged light microscopic and EM datasets. The approach is a significant advance in 3D connectomics as it allows labeled long-range axonal inputs to be mapped onto dense EM neuropil reconstructions. This approach has advantages over other correlative LM-EM approaches as it does not require any chemical label conversion and can potentially be used to study long-range axonal targets of genetically identifiable neurons from selected brain areas at nanometer resolutions afforded by EM. The results presented, especially the matching of varicosities to synapses is very satisfying and convincing. The experimental, computational methods described are sound and well documented. Together the approaches and tools described in this manuscript are likely to be impactful for the connectomics field and this paper merits publication in eLife.

We thank the reviewer for the positive comments.

1) In subsection “Comparison of axon and synapse detection in LM and EM” the authors compare axon and synapse detection in both LM and EM. While it is indisputable that the unambiguous assignment of synapses would require EM data the nearly 30% error rate in the assignment of axonal trajectories and branches in the LM data reported in this study appears unusually high. Is this likely due to high density of viral labeling. It would be useful to understand in more detail the effects of labeling sparsity in LM accuracy. Was the sparsity consistent across the 3 samples studied? Are the sparser regions of the neuropil associated with more accurate tracings?

We agree that this is an important consideration – to address this in more detail, we report the labelling density in both fluorescence channels and report the error rates separately for the two channels (subsection “Comparison of axon and synapse detection in LM and EM"). Furthermore, we tested whether the labelling density variance in the channels (from 0.8 to 1.8% (tdTomato), from 0.7 to 2.3% (eGFP) voxels labelled) along dataset depth correlated to the error rates in reconstruction and found no obvious correlation. Together, this data indicates that at labelling density of 0.7-2.3%, our reported LM-reconstruction error rates are rather consistent. We added a qualifier to the text to make clear that at lower labelling densities, error rates could be lower (subsection “Comparison of axon and synapse detection in LM and EM", Discussion section).

2) The presentation of timing in various parts of the manuscript (Results section, Discussion section and the extrapolation in the Materials and methods section) is a bit confusing and could be better. For instance, what is the relative work load of completion needed for matching versus the time for reconstructing complete axons in the EM volume? It is also not clear if there are other bottleneck steps in scaling to larger volumes and if a linear extrapolation of axon segment matching times is relevant.

We thank the reviewer for pointing this out – we had in fact only reported the time to match, not yet the time for full axon reconstruction (as quantified in Boergens et al., 2017) and control point placement. We have added these times to the respective figure panels (Figure 5E and Figure 6F) and updated the text accordingly (Results section), consistently only reporting the sum of all invested time.

3) The current approach of correlative LM-EM is designed for looking at EM level connectivity of relatively dense axonal projections from different labeled 'source' brain areas. A complementary use case would be to look at input-output relationships of entire axonal arbors of single labeled neurons – the scales of such arbors could be much larger than the EM volume imaged in this study. Could the authors speculate on the feasibility of an approach wherein the acquisition of a high resolution EM dataset would be guided by the LM imaging of a very sparse labeled sample?

This is an interesting suggestion, which we have added as a use case to the

“outlook” figure (new Figure 7C, Discussion section).

Reviewer #3:The manuscript addresses a very important factor in extending the relevance of connectomics for mammalian brains, namely how to assign the source or destination of long-range projections associated with a target neuropil. Presently, the imaging and reconstruction of neural tissue volumes approaching 1 mm^3^ are becoming possible, yet a complete mammalian brain is out of range by 2-3 orders of magnitude. So the identification of long-range input and output connections of a 1 mm^3^ region is clearly a major component to understanding the circuitry a reconstructed local region.The research and manuscript are high quality and comprehensive. As just mentioned the motivation is well justified, the methodology for CLEM is well documented, as well as exemplary data sets and analysis software. The criteria for establishing positive or most likely neuron identification is described thoroughly in both a general level and in the fine detail of source code. The major concerns of error, e.g. error rates from light microscopy reconstruction are presented explicitly and shown how they can still be corrected to give very convincing correlative identification. Several different identification strategies e.g. path length, varicosities, etc. are compared as multiple cross confirming and extending options. Demonstrations in several locations in different layers of the cortex also prove the general applicability of this methodology.

We thank the reviewer for this positive assessment.

One important parameter seems to be the axon labeling density. It seems that this must be controlled to get both as many as possible axons and yet sparse enough that they are separable in LM with the exemplary value of ~71% faithful reconstruction number. It would be helpful to have a rough number here and discussion of how that might be controlled or how that limits or expands the fraction of I/O projections into the EM volume.

We agree – and have added quantifications to the manuscript (subsection “Comparison of axon and synapse detection in LM and EM”). In the presented approach, the fraction of labelled input sources is rather low (about 2% of voxels; assuming at least 70% input axons; which constitute about 50% of the volume in cortex, this would correspond to about 6% of input axons labelled. However, as described in the discussion (Discussion section), one could devise a strategy in which one fluorescence channel is labelled at about 2% sparsity as here, which serves as the alignment channel; and then all other fluorescence channels could be labelled more densely, since co-registration would substantially improve (see Figure 2K and Figure 5H). At the same time, it has to be pointed out that from the perspective of typical fluorescent labelling experiments, our labelling rate is already rather high, as also judged by reviewer 2.

There is without doubt a highly relevant, timely, and useful contribution to the field.